# A Study on the Origin of China's Modern Industrial Architecture and Its Development Strategies of Industrial Tourism

**Rui Han [1,2,3,*], Daping Liu [1,*] and Paolo Cornaglia [3]**

[1] School of Architecture, Key Laboratory of Cold Region Urban and Rural Human Settlement Environment Science and Technology of Ministry of Industry and Information Technology, Harbin Institute of Technology, Harbin 150006, China

[2] College of Art and Design, Creative Center for ArtSciArch, Jilin Jianzhu University, Changchun 130118, China

[3] Department of Architecture and Design, Politecnico di Torino, 10125 Torino, Italy; paolo.cornaglia2@gmail.com

[*] Correspondence: rui.han@polito.it (R.H.); ldp_abc@hit.edu.cn (D.L.); Tel.: +86-186-4319-9988 (R.H.)

**Abstract:** Due to the unique cultural attribution and facade aesthetics, China's modern industrial architecture, built in the 1950s, played a significant role and expressed a specific historical value in the process of human industrial civilization. The objective of this study was to reveal the origin of China's modern industrial architecture, meanwhile understanding the content, the channel, and the process of the global transfer of modern industrial architecture from the United States to the Soviet Union and then to China. With a literature review, we summarized the United States' achievements of modern industrial architecture at the beginning of the 20th century and described the formation and evolution process of the Soviet Union's modern industrial architecture from the 1920s to the 1950s. Through field investigation and measurement into China's modern industrial plants, we comparatively analyzed the inheritance and changes among the United States, the Soviet Union, and China from the perspective of the planning concept, design theory, and structural technology. Finally, two sustainable development strategies of industrial tourism were proposed for China's modern industrial heritage according to the comprehensive assessment, and two typical development patterns were presented based on their respective advantages.

**Keywords:** modern industrial architecture; global transfer; design theory; industrial tourism; sustainable development

## 1. Introduction

In the first half of the 20th century, the social circumstances were turbulent all over the world. There were wars and revolutions, exclusion and fusion, development and innovation. On the one hand, this caused great injuries and challenges to human society; on the other hand, it promoted the rapid development of the economy, the great progress of science and technology, and the integration of different cultures. Among the positive consequences, the global transfer of modern industrial architecture is particularly striking.

At the end of 18th century, from the birthplace of industrial civilization, the European continent, industrial technology was exported to the United States. Following the rapid economic development after the American Civil War (1861–1865), the industrial architecture and construction technology became increasingly mature [1]. In the 1930s, due to the turmoil of the European political environment, the flow of architects among European countries became more frequent, which promoted the exchange

and innovation of industrial architecture and benefited the development of modernist architecture [2]. Meanwhile, with the spread of the global economic crisis (1929–1933) among capitalist countries, exporting industrial technology became the panacea to alleviate the shortage of domestic consumption in the United States. Inevitably, the Soviet Union, which was still in the junior period of modern industrial formation, became the main destination to receive the achievements of the mature West in industrial architectural theory and technology.

During the Soviet Union's First Five-Year Plan (1928–1932), the United States and other European countries provided industrial assistance to help the Soviet Union establish a modern industrial system. In this process, modern industrial architecture, with new structural technology, was imported into the Soviet Union. Then, the Soviet Union carried out large-scale industrialization, from the Second Five-Year Plan to the Fifth Five-Year Plan (1933–1954), becoming a strong modern industrial and technological country. An abundant innovative theoretical and practical experience was gradually accumulated and began to be exported to its leading allies in the form of industrial assistance, which inevitably carried the industrial standard and political ideology [3]. In the 1950s, China signed the "156 Projects" assistance contract with the Soviet Union. Modern industrial architecture began to be completely imported into China from the Soviet Union during China's First Five-Year Plan (1953–1957). To meet the needs of Chinese political culture and express the prevailing ideological trend of architectural design, the innovation of modern industrial architecture emerged. China's modern industrial architecture and industrial standard were gradually formed, which became the foundation of China's future industry [4].

Based on a literature review and field investigation, we described the industrial transfer route from the United States to the Soviet Union and then to China in the first half of the 20th century, revealing the transfer process of the modern industrial architecture planning concept, design theory, and structural technology. This process was full of inheritance and changes, exclusion and integration, radicalization and retrogression, which became a scarce and precious example of the process of human industrial civilization. Besides, this study inspired the consideration of sustainable development strategies for China's modern industrial tourism in the future. The global transfer route of the modern industrial architecture is shown in Figure 1.

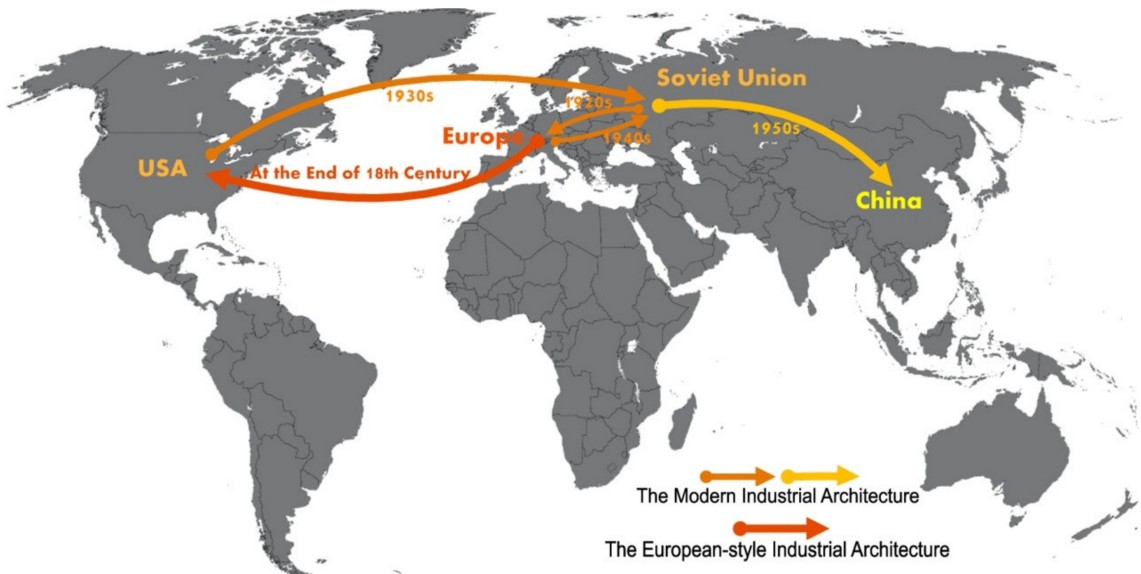

**Figure 1.** The global transfer route of the modern industrial architecture.

## 2. Background

### *2.1. The Development of Modern Industrial Architecture in the United States*

At the end of 18th century, industrial architecture and technology flowed into the United States with immigrants and took root in the vast land of the New World. At that time, the industrial development in the United States maintained the same pace as that in Europe.

The design of industrial buildings was completely due to European education. This is evidenced by the following features of industrial buildings: (1) A single workshop included a factory; (2) a masonry structure was generally applied to a workshop; and (3) there was lots of decoration on the facade. Industrial buildings with above features were called European-style by the Americans. In the second half of the 19th century, the economic reconstruction after the American Civil War (1861–65) received more and more investment, which caused the revolution of new technology and new materials in the industry. The United States gradually took the leadership of the modern industrial development. The explosion of the automobile industry had brought great opportunities for the modern industrial architecture in the United States. More and more outstanding American industrial architects began to emerge on the stage of history and tried their best to respond to the mission call of the great era with their talent and diligence. They created an advanced modern industrial architectural design theory and invented new structural technology. Albert Kahn was the brightest and most influential representative of these American architects. He was called the father of modern industrial architecture due to his great contribution to the development of modern industrial architecture in the United States, European countries, and even the Soviet Union [5].

### 2.1.1. Albert Kahn's Planning Concept of the Modern Industrial Plant

Albert Kahn was a stubborn functional supremacist who advocated that all planning and design should serve industrial manufacturing as much as possible. In his design project of the Pierce Plant (1906), he put a lot of energy into the exploration and expression of the functional supremacy of modern industrial architecture and translated all his concepts and theories into plant planning and workshop design [6]. In order to meet the requirements of the manufacturing process, he arranged six main workshops in an east–west orientation and integrated the garage, the welding workshop, and the power station into a group in the north of the plant, which made the northern railway line serve the group conveniently. This planning ensured that the materials and products were transferred in the north–south orientation, while the assembly line and manufacturing process were transferred in the east–west orientation. The three main workshops—the manufacturing workshop, assembly workshop and car body workshop—were integrated and designed as a comprehensive huge workshop, which greatly improved the manufacturing efficiency. The facade of the huge workshop was so concise that large-area windows were used to guarantee day lighting [7]. The planning of the Pierce Plant laid several principle foundations for modern industrial architecture: (1) The plant should be arranged into several groups according to their different function; (2) the transfer directions of the manufacturing process and materials should be separated; and (3) the huge workshop can meet the needs of a longer assembly line. The general planning and the workshop's facade are shown in Figures 2 and 3, respectively.

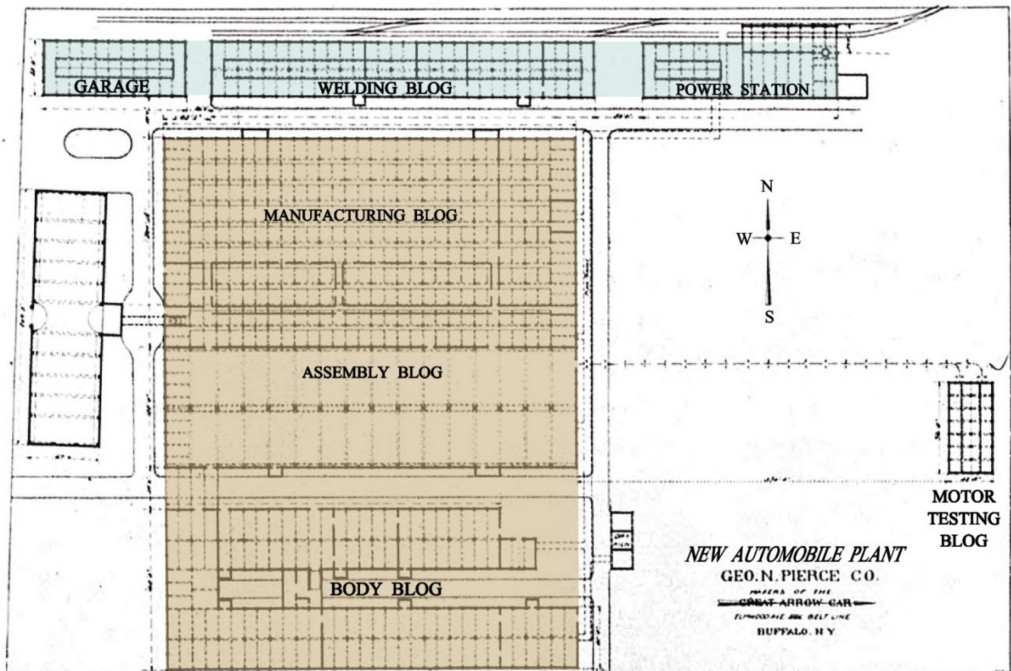

**Figure 2.** The general planning of the Pierce Plant. (Source: [8]).

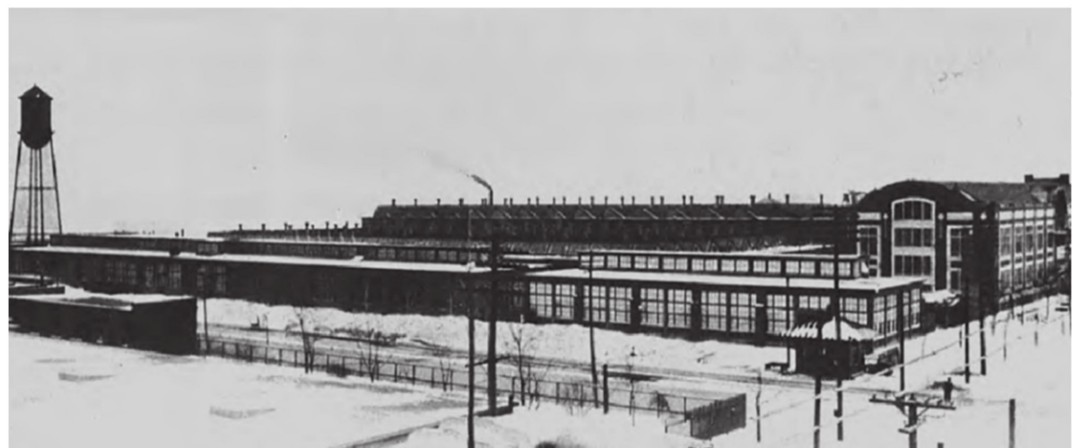

**Figure 3.** The workshop facade of the Pierce Plant. (Source: [8]).

Albert Kahn's planning concept was widely applied in all his industrial projects. The Ford Rouge Complex designed by Albert Kahn in 1917, had a more complicated planning layout. It used to be the largest comprehensive industrial base in the world. The planning had such a lucid logic that it presented reasonable functional workshop groups and a clear traffic flow. Albert Kahn paid more attention to the organization of the manufacturing process and reserved free construction land for future development, finally realizing sustainable spatial planning for the modern industry [9]. The Ford Rouge Complex had a profound influence, which directly inspired the Gorkovsky Avtomobilny Zavod, founded by the Ford Motor Company in the 1930s as industrial assistance for the Soviet Union. Then, the Soviet Union's architects copied the planning and design to help China establish the Changchun First Automobile Works, which was the core plant of "156 Projects" [10]. Thus, the indirect influence of American modern industrial architecture on China can be observed. Figure 4 shows the planning of the Ford Rouge Complex.

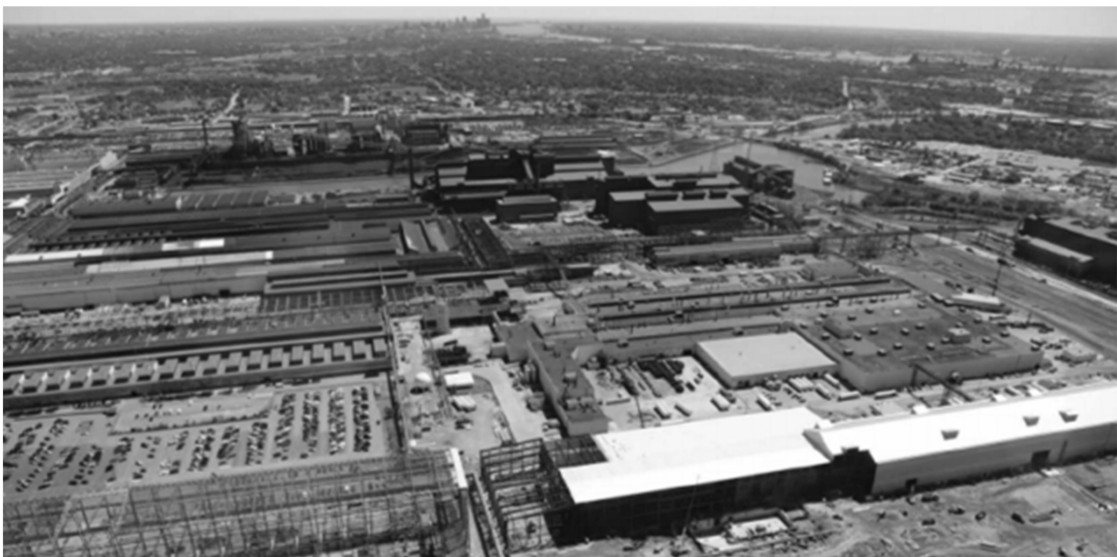

**Figure 4.** The planning of the Ford Rouge Complex. (Source: [11]).

2.1.2. Albert Kahn's Exploration of Structural Technology

The American Civil War intensified the conflict between modern industrialization and the traditional handicraft industry. The application of new materials and technologies, which were born of the modern industrialization system, brought military advantages to the victorious side of the American Civil War, and even further the development of modern engineering technology. Albert Kahn focused on researching not only the new structural technology but also its application in design works. He proposed that functional needs play a dominant role in the design of the facade, the internal space, and the external form of a modern workshop. This theory also included the idea that the structure should serve the manufacturing process and arrangement of the assembly line [12]. The early industrial architecture, transferred from Europe to the United States, widely applied a multi-story workshop design and load-bearing wall structure, which seriously limited the size of the internal space and the position of the windows on the facade. Accordingly, day lighting and ventilation were very poor. Albert Kahn led the innovation of the structure of the American modern industrial architecture by improving and popularizing the reinforced concrete frame structure and steel structure frame [8].

In 1903, the No. 10 workshop of the Packard Automobile Plant, designed by Albert Kahn, became the first automobile workshop to use a reinforced concrete frame structure in American industrial history. The internal space span was up to 9.8 m, which realized a large-span internal space that was more conducive to the organization of manufacturing [13]. While meeting the functional requirements of arranging the assembly line, the structure greatly enhanced fire resistance and stability. The facade of the No. 10 workshop of the Packard Automobile Plant is shown in Figure 5.

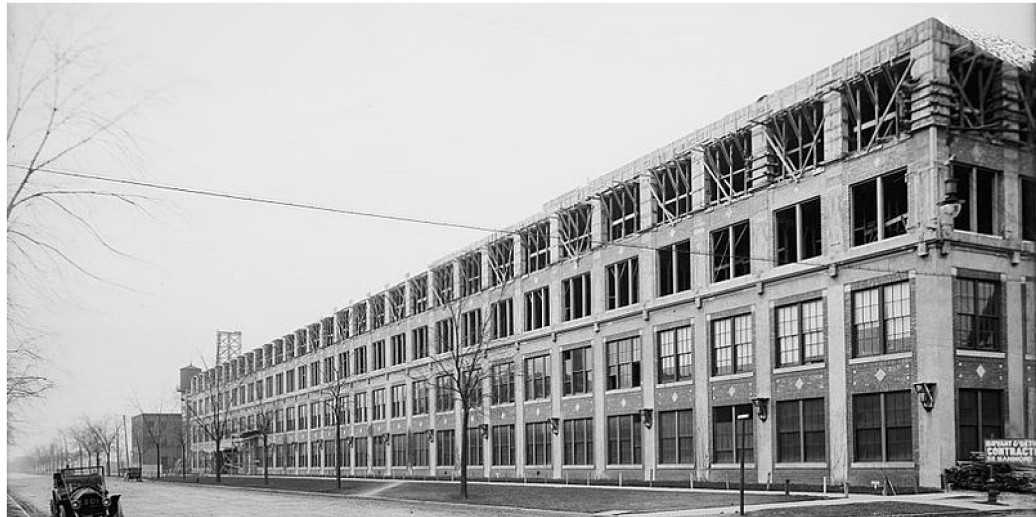

**Figure 5.** The facade of the No. 10 workshop of the Packard Automobile Plant. (Source: [14]).

The reinforced concrete frame structure made the centralized layout of a huge workshop a reality. The multi-oriented and openable roof windows became the key to solving the problem of insufficient lighting and ventilation in a huge workshop. In 1906, Albert Kahn used the continuous zigzag roof windows in the design of the Pierce Plant, which made the schema scale of the modern industrial workshop no longer limited and more convenient for arranging the assembly line [15]. The external form of the Pierce Plant workshop is shown in Figure 6.

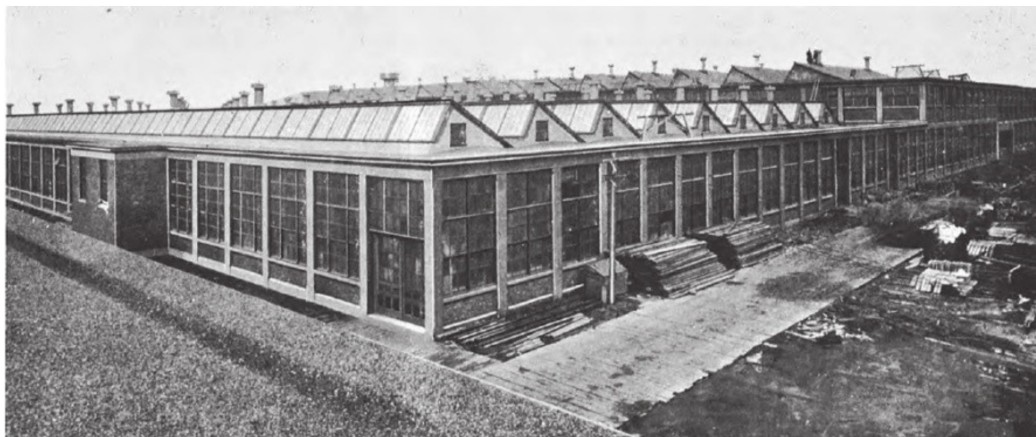

**Figure 6.** The continuous zigzag roof windows of the Pierce Plant workshop. (Source: [8]).

The innovation of the modern industrial workshop structure promoted the progress of the manufacturing process. In 1908, Albert Kahn designed a four-story automobile manufacturing workshop for the Ford High Land Plant. This workshop completely applied Kahn's reinforced concrete frame system, invented by Albert Kahn's brother, Julius Kahn. The floor and beam of the workshop were poured as a whole body. The continuous main beam span reached 7.6 m. This structure helped the workshop form a wide and unobstructed internal space, which provided the best space for the automobile assembly line [16]. After the completion of this workshop, it immediately became a popular star in the United States and Europe, and attracted lots of industrial architects, who came to learn from it. It was eventually considered as a significant milestone in the development of modern industrial architecture. It promoted the rise of modern industrial architecture in Europe and directly inspired the Lingotto Fiat Factory in Torino, designed by the Italian engineer, Mattè-Trucco,

in 1922 [17]. The workshop of the Ford High Land Plant and Lingotto Fiat Factory are shown in Figures 7 and 8, respectively.

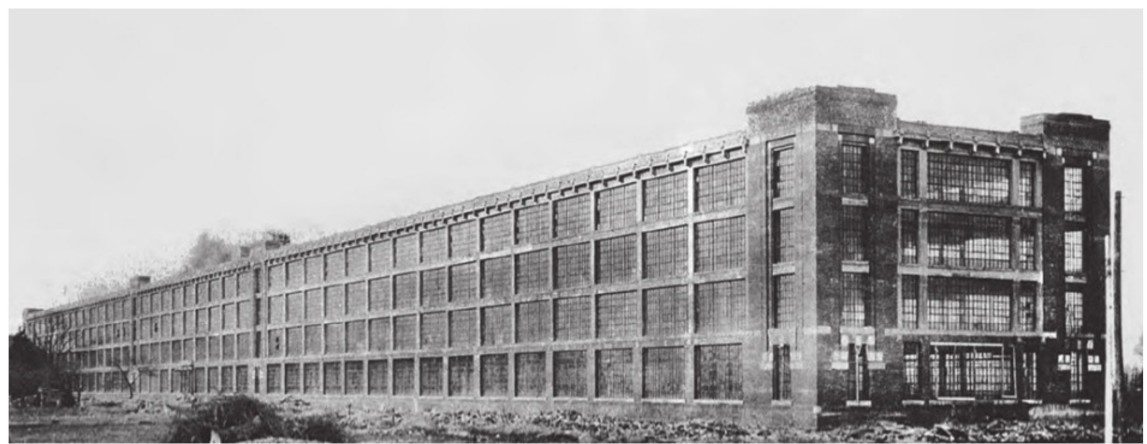

**Figure 7.** The workshop of the Ford High Land Plant. (Source: [8]).

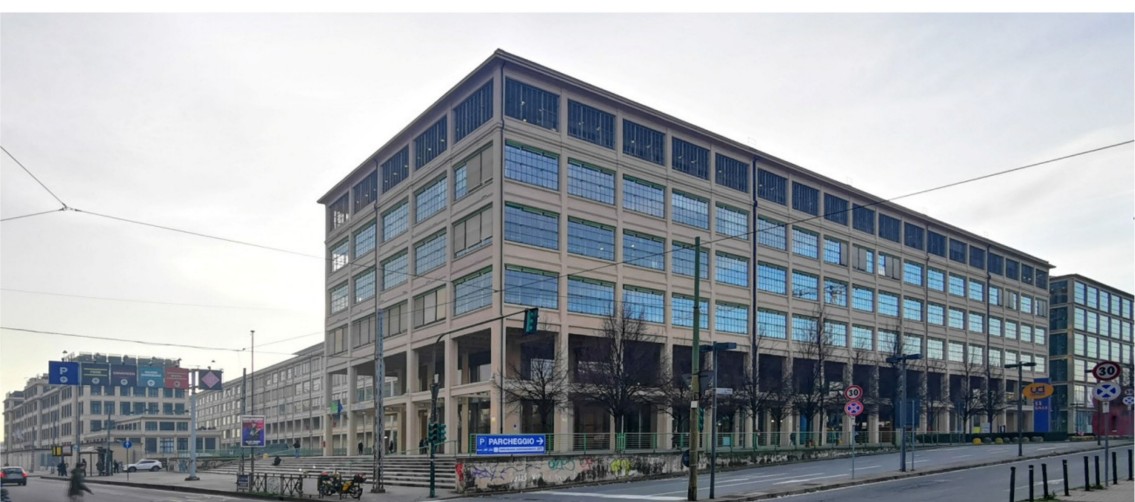

**Figure 8.** Lingotto Fiat Factory.

In order to further improve the internal space utilization of modern industrial workshops and meet the needs of different manufacturing process, Albert Kahn began to focus on the exploration of the steel frame structure. He summed up his experience in a large number of project practices, which helped him design a series of steel frame structures to realize a close connection and high integrity for the manufacturing process. Albert Kahn indicated that the structural design of modern workshops should follow a principle of simplification. The steel frame structure can meet this principle due to its advantages, which are as follows: (1) A clear node logic; (2) a simple installation and operation; (3) a reduction in labor costs; and (4) an improvement in construction efficiency. The forging workshop (1911) of the Packard Plant and the glass workshop (1922) of the Ford Rouge Complex were the best cases of the application of Albert Kahn's structural design theory. Due to the optimization of the steel structural systems, the internal space of these two workshops could serve the manufacturing process better. In addition, the steel structural systems supported larger windows on both the facade and the roof, which brought better day lighting and ventilation, as well as making the external form of the workshops full of geometric aesthetics and visual tension [18]. The forging workshop of the Packard Plant and the glass workshop of the Ford Rouge Complex are shown in Figures 9 and 10, respectively.

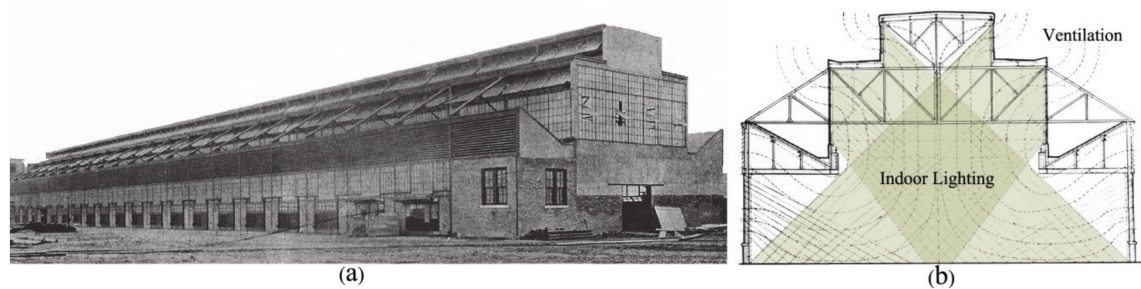

**Figure 9.** The forging workshop of the Packard Plant: (**a**) the facade and (**b**) the section. (Source: [8]).

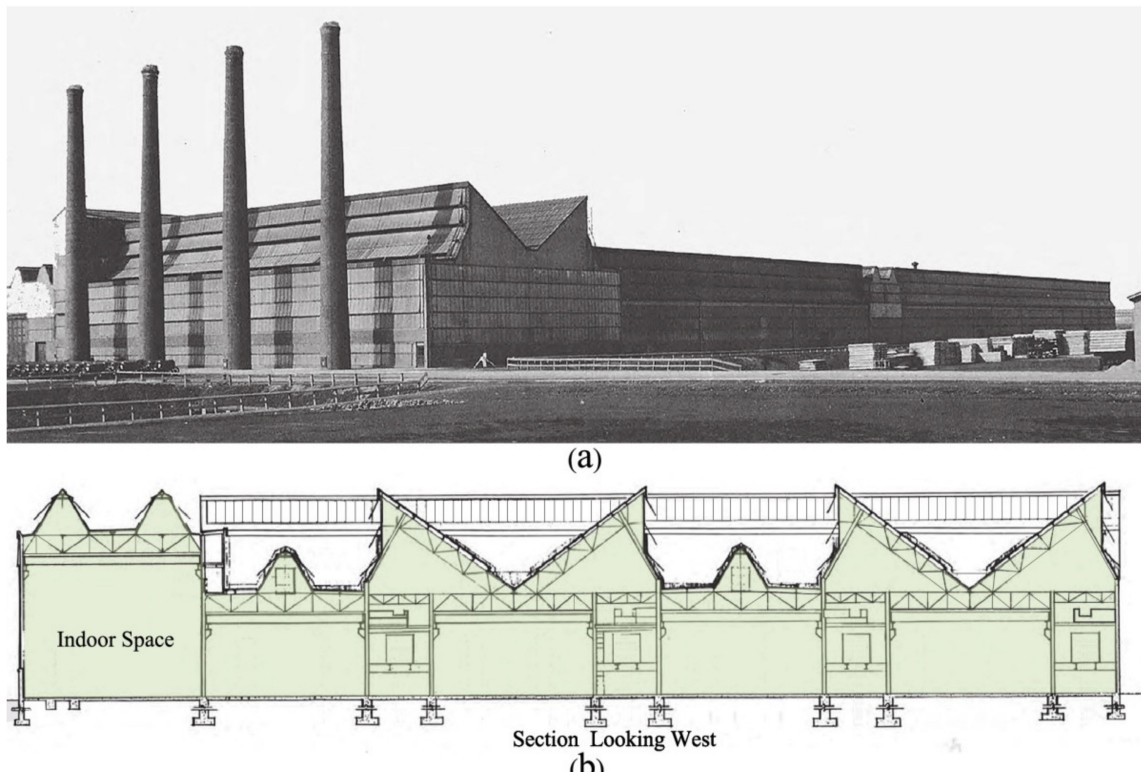

**Figure 10.** The glass workshop of the Ford Rouge Complex: (**a**) the facade and (**b**) the section. (Source: [8]).

### 2.1.3. Albert Kahn's Design Theory of the Modern Industrial Workshop

With the help of a reinforced concrete frame structure and steel frame structure system, the position and the size of the windows were no longer limited, which promoted the use of large-area windows on both the facade and the roof. The modern industrial building aesthetics began casting off the shackles of classical architectural aesthetics and created a new and unique aesthetic system. For example, the forging workshop of the Packard Plant, with a single-layer steel frame structure, had formed an open large space, since 52% of the windows on the facade could be opened, and 70% of the plan area could achieve natural lighting [19]. Sufficient day lighting and ventilation, a clean and beautiful internal space created a comfortable and pleasant working environment for workers. The darkness, congestion, and depression of the traditional workshops were gradually abandoned.

Albert Kahn contributed a lot to the development of modern industrial architecture. He defined the basic design theory of it, which makes us establish a clear understanding of modern industrial architecture. Albert Kahn's design theory is summarized as follows: (1) The function determines the schema size and facade decoration of the workshop; (2) the arrangement of the column net and the selection of the structural system are determined according to the manufacturing process; (3) the order

and the division of the windows enhance the rhythmic sense of the facade aesthetic; (4) the external form of the workshop is simplified in order to present the characteristics of structural and functional aesthetics; and (5) day lighting and ventilation are the basis of a comfortable workshop.

2.1.4. The Transfer of Modern Industrial Architecture from the United States to the Soviet Union

After the October Revolution (1917), the Soviet Union began its unprecedented industrial construction, after a short-term economic recovery. The Soviet Union began to implement its First Five-Year Plan (1928–1932) and gave priority to the development of the industry. At the same time, the world economic crisis spread rapidly from the United States to the whole capitalist world in 1929. The economic decline obviously influenced industry. The United States expected to alleviate the domestic economic crisis through industrial export, which satisfied the desire of the Soviet Union to attract industrial assistance [20]. In May 1929, The Ford Motor Company was hired to provide industrial assistance services and training services by the Soviet Union's government. A large number of Ford's employees were sent to work in the Soviet Union until 1933. Albert Kahn was hired as a senior consultant because of his outstanding fame and excellent talent in the field of modern industrial architecture. He led 25% of the employees of his firm to Moscow and directed a large number of industrial projects in the Soviet Union. He also introduced and exported his structural technology of the reinforced concrete frame and steel frame, which helped the Soviet Union establish its structural standard [21].

The Chelyabinsk Tractor Plant, built in 1933, was one of Albert Kahn's most representative design projects in the Soviet Union. The main workshop span was up to 30.5 m, with a height of 12.2 m. This workshop brought lots of changes to the manufacturing process and greatly improved the assembly line efficiency. It drew on the design experience of the glass workshop of the Ford Rouge Complex. Albert Kahn made full use of all his theories. Firstly, he researched on the manufacturing process of the tractor and then decided to use a steel frame structure to design a huge workshop for a multi-function and complicated assembly line. Secondly, he designed large-area windows on the facade and the roof, which provided good day lighting and ventilation. Finally, he finished designing this workshop's external form with a futuristic sense of geometric aesthetics [22]. The external form and the section of the main workshop of the Chelyabinsk Tractor Plant are shown in Figures 11 and 12, respectively.

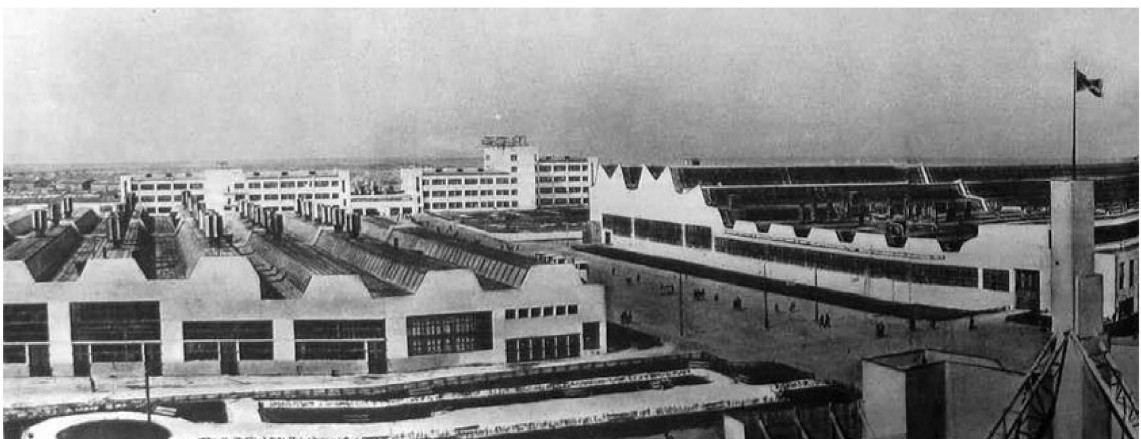

**Figure 11.** The external form of the huge workshop of the Chelyabinsk Tractor Plant. (Source: [22]).

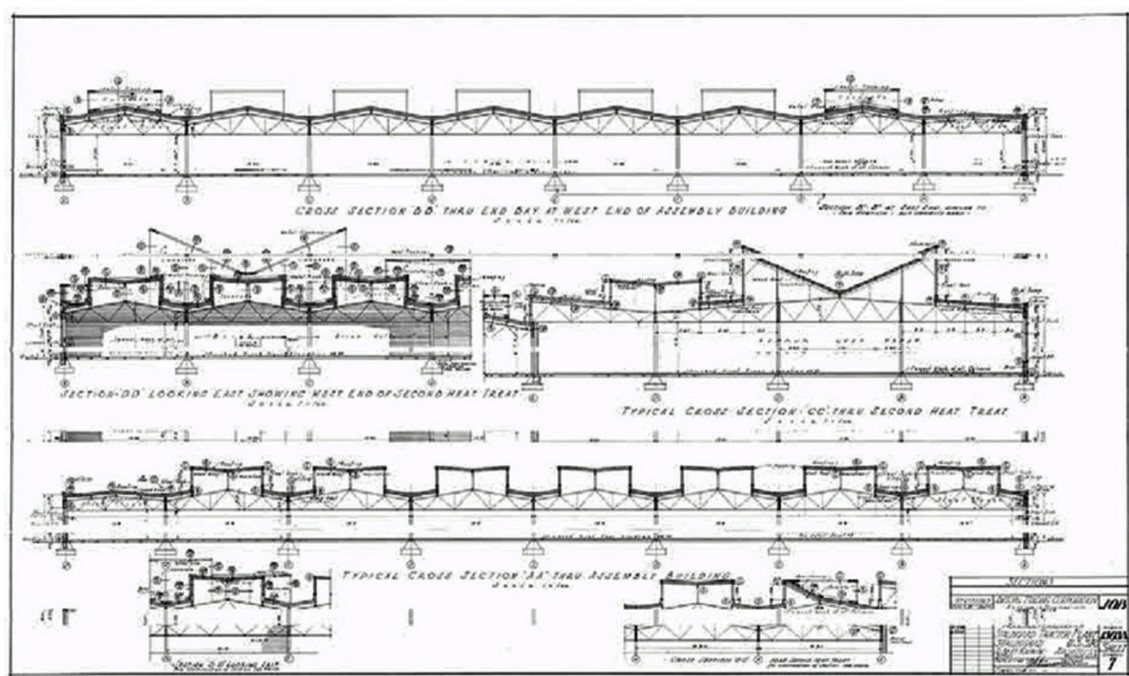

**Figure 12.** The section of the huge workshop of the Chelyabinsk Tractor Plant. (Source: [22]).

While directing and designing 531 industrial plant projects, Albert Kahn trained more than 4000 Soviet architects and engineers, and introduced his planning concept, design theory, and structural technology in three years. These architects and engineers later grew up and became the backbone of the Soviet Union's industrial architecture [23]. Albert Kahn helped the Soviet Union quickly absorb the successful experience and technical achievements of the United States' modern industrial architecture.

### 2.2. The Development of Modern Industrial Architecture in the Soviet Union

#### 2.2.1. The Influence of Constructivism on the Soviet Union's Industrial Architecture

The Soviet Union was in the stage of wartime communism from 1918 to 1921. Lenin, the leader of the state, held an open and encouraging attitude towards artistic creation. After constructivism had been invented by the Soviet Union's artists, the architects began to explore how to interpret it into architecture. At the beginning of the implementation of Lenin's new economic policy in 1921, the Soviet Union still maintained positive contact with the Western world. A large number of economic and cultural interactions took place between the governments and people, which provided a good environment for the development of constructivist architecture [24].

From 1923 to 1924, an architectural group, with A. Vesnin as the core, was established in the Moscow Institute of Art and Culture. Then, this group established the Association of Contemporary Architects in 1925. A. Vesnin served as the chairman and leader. He proposed that technology and structure should provide a method for solving the problem of architectural functions, and all the buildings should be generated from inside to outside. At the same time, there also existed the Association of New Architects, which was very active and influential. It was established by N. Ladvosky in 1923. He proposed that psychological power was the main criterion for the formation of architectural forms, and the process of architectural design had the objective of solving the problem of how to organize space [25]. At that time, the Soviet Union's architects maintained frequent contact with German and French architects. The theory of constructivism even had a positive impact on modernist architects, such as Walter Gropius and Le Corbusier [26].

The architects who took part in the exploration of constructivism in the Soviet Union concluded that the spatial structure of architecture was the starting point of architectural design. This theory

gradually became the basic principle of world modernist architecture and was eventually suitable for the original idea of modern industrial architecture with the feature of functional supremacy [27]. Before importing Albert Kahn's theory of modern industrial architecture, constructivism had become the main style of modern industrial architecture in the Soviet Union.

### 2.2.2. The Influence of Socialist Realism on the Soviet Union's Industrial Architecture

The Soviet Union carried out the First Five-Year Plan (1928–1932) and achieved great success, which led to the centralization of politics, culture, and the economy. Stalin, the leader of the state, began to strengthen the control of social culture and ideology [28]. In April 1932, the Central Committee of the Soviet Union launched the documents of the Reorganization of Literary and Art Groups, which forced all the writers and artists to join a new association under the leadership of the Bolshevik Party. The socialist realism theory was established as the basis of the Soviet Union's literary creation and soon became the basic theory of all the artistic creation, such as painting, sculpture, film, and even architecture [29].

From the 1930s to World War II, the political environment in the Soviet Union began to cause regress in architecture, reverting back to a worship of classicism. The architectural style was driven back to the Russian Imperial style, invented in Moscow's urban reconstruction in the second half of the 19th century [30]. The interpretation of the socialist realism theory was pushed into the abysm of aspiring humanism and Utopia. Socialist realism was eventually transformed into a populist and expressionist ideology, which led to an architectural degeneration to the classical revival style [31]. With the continuous practice and summary of the socialist realism style in the field of civil architecture, the methodology, called the national form plus the socialist content were presented and gradually matured [32]. The expression of socialist content was described the people's longing for socialism and a new social life. It was presented in the first conference of Soviet writers in 1931. The socialist content commanded that architecture should meet not only the requirements for creating suitable material space, but also the people's need for ideology and culture [33]. The socialist content was too abstract to find a specific representation in architectural discourses. However, through a large number of case studies, we found that it mainly emphasized a clear central axis and symmetrical layout in the general planning and facade design. The national form of facade was described to express the Russian Imperial style, which was composed of the Baroque style, Renaissance style, and Gothic style. There are three features: (1) A large number of complex decorations and graphics; (2) a remodeling of the Russian traditional tent roof; and (3) a wide application of a three-stage classical facade [34].

### 2.2.3. The Selection of Modern Industrial Architecture in the Soviet Union

Rapid industrialization led to a structural change of the urban population. A large number of rural people entered the cities and towns. Their aesthetic taste inevitably had an impact on architecture. The constructivist architecture was considered too rational and callous by these new urban residents. In addition, the government preferred to use a traditional architectural style to unite people's spirit. As a result, the socialist realism theory dominated in the field of civil architecture, for example, in important monumental buildings, military and political office buildings, cultural exhibition buildings, and residential buildings. However, constructivism still took the leading position in industrial architecture and eventually became the main planning and design methodology of modern industrial architecture in the Soviet Union [35]. While the national form plus the socialist content became the representation of the national ideology and the main export carrier of politics and culture, constructivism was always the main theme in the field of industrial architecture. The reasons are as follows: (1) The original industrial architecture of the Soviet Union was established by domestic constructivist architects; (2) the Soviet Union absorbed the theory and technology of modern industrial architecture from the United States, which were consistent with constructivism concept; and (3) in order to realize large-scale modern industrial construction, the Soviet Union invented a standardized design and prefabricated concrete

technology. As a result, the simplified decoration and functional supremacy, which constituted the main content of constructivism, became widely accepted [36].

2.2.4. The Development of Modern Industrial Architecture (1917–1932)

The modern industrial architecture in the Soviet Union started from the planning and design of power plants. After the October Revolution (1917), in order to rebuild the national economic system, the government carried out a large-scale construction of power plants. With the help of these power plants, a sufficient energy supply was guaranteed. As a result, more than 300 large-scale industrial plants and thousands of industrial workshops were built in the Soviet Union from 1917 to 1928 [37]. The Soviet Union had achieved a lot of experience in the planning and design of modern power plants, and successively built the Kashmirsk Power Plant (1922), Kizelovsk Power Plant (1924), Shatura Hydropower Station (1925), Sverdlovsk Power Plant (1926), Volkov Hydropower Station (1926), Yerevan Power Plant (1926), Strovsk Hydropower Station (1928), Ivanovo Wozneskgris Hydropower Station (1928), Leninnakan Hydropower Station (1928), and Armenia Kondobozskaya Power Plant (1928) [38]. All of these power plants were in a constructivist style.

The Soviet Union's First Five-Year Plan (1928–1932) was the peak of modern industrial architecture development. The Dnieper Hydroelectric Station, a huge hydropower station, was planned. Many excellent architects and engineers took part in the design and construction from 1927 to 1932. The American engineer, A. Winter, served as the chief engineer. The design team was led by the Soviet Union's architect V. Vesnin. The American architect, H. Cooper, was hired as the senior consultant. The Dnieper Hydroelectric Station, with its installed capacity of 558,000 KW, became the largest hydropower station in the world at that time. The dam was made of concrete, with a series of large vertical wedge-shaped bodies, which made sense of strength and power. The workshop of the power machine was located at the end of the dam, and was 250 m long and 20 m wide [39]. V. Vesnin used a large number of windows on the facade to provide the workshop with better day lighting and ventilation. The staff could observe the water level outside of the windows. Both the dam and the workshop fully used modern industrial architecture and constructivist theory. The dam of the Dnieper Hydroelectric Station is shown in Figure 13. The facade and internal space of the workshop are shown in Figure 14.

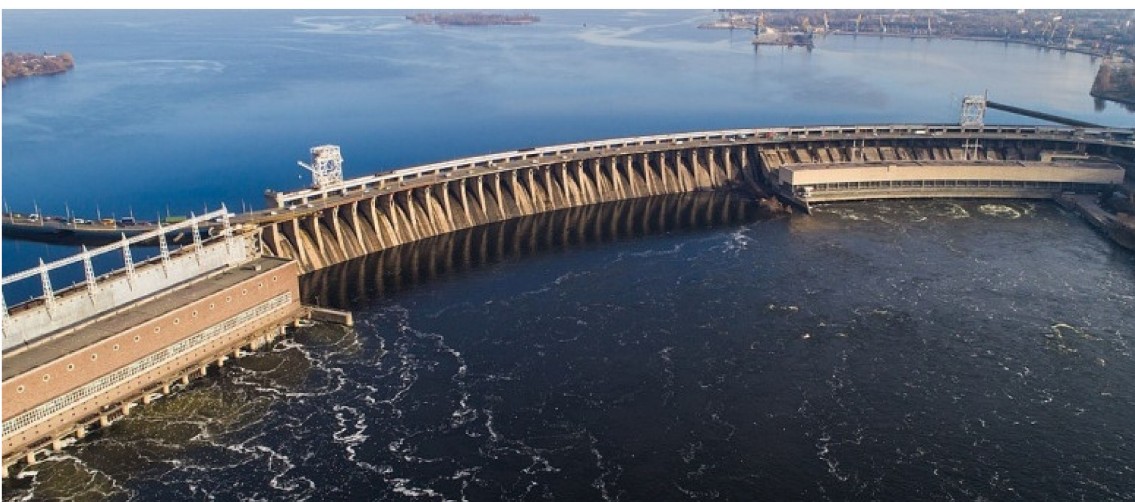

**Figure 13.** The dam of the Dnieper Hydroelectric Station. (Source: [40]).

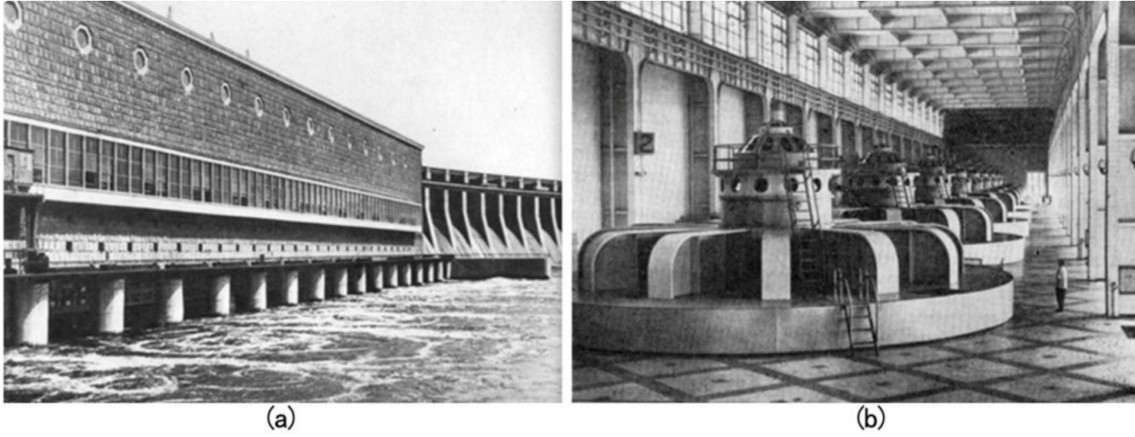

**Figure 14.** The workshop of the Dnieper Hydroelectric Station: (**a**) the facade and (**b**) the internal space. (Source: [38]).

The industrial architecture of the light industry had also made considerable progress from 1928 to 1932. It fully absorbed the experience of the architectural design of heavy industry in the Soviet Union and learned a lot from the United States. The features of simplicity and functional supremacy were expressed. A number of light industrial plants and workshops, with a high design level, emerged. The Ivandevka Knitting Factory, designed by G. Goltz, I. Sobolev, and M. Parusnikov, was built near Moscow in 1930. The main workshop was a six-story building, with a reinforced concrete frame structure. Despite the external form of the workshop was quite concise, the proportion was controlled appropriately. The big window curtain wall with rich details formed a strong contrast with the large area of the blank envelope wall. The architects used two stairwells to divide the facade geometrically, which allowed the workshop to be full of proportional and harmonious vision. They also innovatively designed a garden in front of the workshop, which realized two goals: (1) to allow the road to be divided by a garden and separate the people flow from the vehicle flow; and (2) to enrich the natural environment and make workers relax physically and mentally [41]. Figure 15 shows the Ivandevka Knitting Factory.

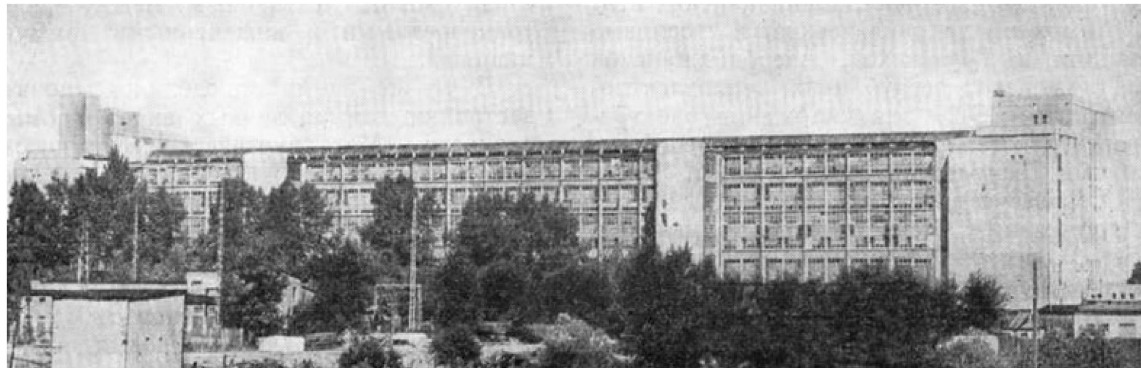

**Figure 15.** The Ivandevka Knitting Factory. (Source: [38]).

The transfer of modern industrial architecture from the United States to the Soviet Union played a very important role in the development of the Soviet Union's industry in the First Five-Year Plan. The American Austin Company and Ford Motor Company helped the Soviet Union establish the foundation of its modern industry. In this process, the American architect, Albert Kahn, led other architects and engineers to introduce the mature theory of modern industrial architecture and structural technology into the Soviet Union through directing projects and training the Soviet Union's architects. This process was facilitated by the commonality that existed between modern industrial architecture

and constructivism in their concepts and methodologies. Meanwhile, the reinforced concrete structure and steel frame structure, imported from the United States, also promoted the development of modern structural technology in the Soviet Union [42]. In the middle of the First Five-Year Plan, the Soviet Union began to explore a standardized design for the purpose of large-scale industrial construction. The standard of a column net of 3 m, as an expansion modulus, was determined. As column net standards in industrial workshops, 6 m, 9 m, and 18 m were widely used. The standardized design was effective and valuable for controlling the spatial planning of industrial plants [43]. At the end of the First Five-Year Plan, the Soviet Union began to use prefabricated concrete components, such as beams, columns, roof slabs and stairs, in the construction of some workshops, which promoted the development of structural standardization and architectural industrialization in the Soviet Union.

2.2.5. The Development of Modern Industrial Architecture in the Second Five-Year Plan and Third Five-Year Plan (1933–1941)

Constructivism was still the main theme in the field of industrial architecture in the Second Five-Year Plan (1933–1937) and Third Five-Year Plan (1938–1941). The use of stained-glass windows made the facade full of bright and pleasant vision in this period. The Kramatorsk Plant was built with a prefabricated reinforced concrete structure in 1936. The facade of the workshop was expressive. The large-scale stained-glass curtain wall and the red brick envelope walls on both sides made the flat facade present a geometric and proportional vision. The internal space also achieved better lighting [44]. The workshop of the Kramatorsk Plant is shown in Figure 16.

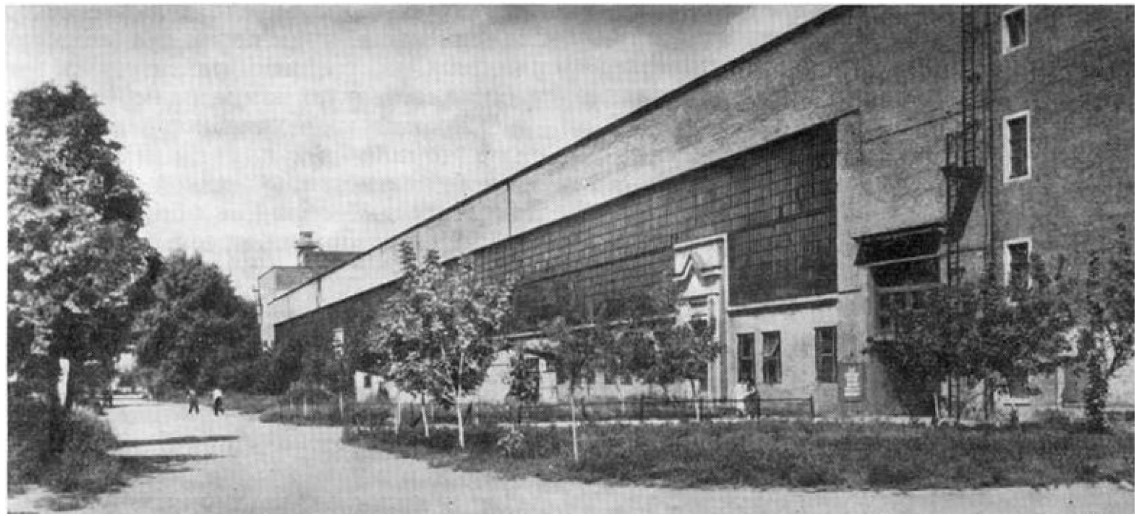

**Figure 16.** The workshop of the Kramatorsk Plant. (Source: [45]).

A kind of new industrial building that integrated the workshop and office building emerged in this period. It usually had multi-story, complex functional zones inside, and a better arrangement of horizontal traffic and vertical traffic, which brought challenges to the Soviet Union's architects and structural engineers. The Moscow Stalin Automobile Plant, designed by E. Popov and S. Muravyo, in 1935 applied a standardized design and 6 m × 12 m column net to integrate the office area and workshop area successfully. The architects used a color and shape discourse to distinguish different functional zones. In addition, they also paid a lot of attention to the garden design in the plant [45]. The new industrial building of the Moscow Stalin Automobile Plant is shown in Figure 17.

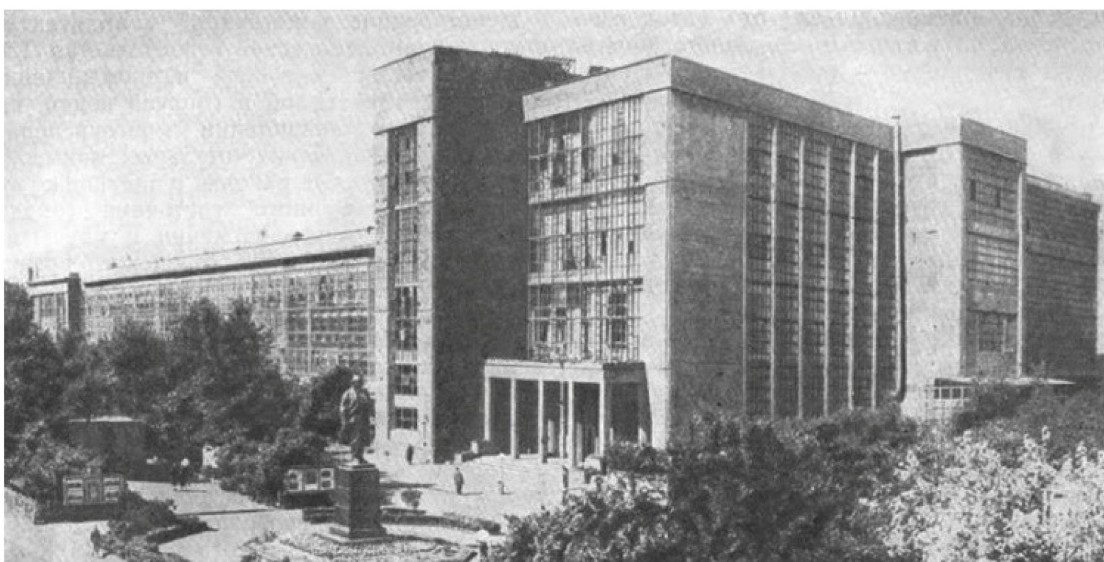

**Figure 17.** The new industrial building of the Moscow Stalin Automobile Plant. (Source: [45]).

In the Second Five-Year Plan, the increasing production of the steel industry and cement industry promoted the development of long-span structural technology and the emergence of workshops with a larger centralized layout. The number of workshops with prefabricated concrete columns and a steel roof truss was increasing. As a result, the architects could design larger areas of glass curtain walls to realize better lighting and ventilation. In the Third Five-Year Plan, the Soviet Union established a large number of industrial design schools. A lot of architects and engineers were trained, which laid the foundation for the future export of industrial architecture.

### 2.2.6. The Development of Modern Industrial Architecture in the Fourth Five-Year Plan and Fifth Five-Year Plan (1946–1954)

During World War II (1939–1945), thousands of industrial plants in the Soviet Union were destroyed. The Fourth Five-Year Plan began to be implemented in 1946. The recovery and rebuilding of industry became the primary task. The government launched a strict economic simplified construction standard to control the cost in the event of a serious lack of steel and cement. As a result, the structural standard of the industrial workshops was also changed. Column nets of 6 × 18 m, 12 × 15 m, and 12 × 18 m gradually took the place of the original standard column net of 6 m × 12 m. The width of all new workshops was controlled at 48–60 m [46].

More and more huge industrial bases appeared in this period. They usually consisted of several big industrial plants. A large number of workshops, auxiliary buildings and living areas, with more complex functions, were superimposed on each other. The Soviet Union's architects had to use new concepts and methods to solve new problems, which stimulated the improvement of urban planning and design level in the Soviet Union [47].

In spite of having different ideologies, the Soviet Union still focused on learning and importing advanced construction technology from the other European countries [48]. At that time, Britain and France made great progress in precast concrete technology and invented more economical and reasonable precast technologies, which greatly improved the construction efficiency and reduced the construction cost [49]. The government of the Soviet Union began to send engineers to visit Britain and France. They brought new technology back. More and more precast concrete beams, precast concrete columns and precast concrete floors were widely used in industrial plants and even in power plants. The Moscow No. 12 Power Plant used a lot of precast components, which realized a shorter construction period. A novel design concept of industrial architecture appeared to accompany the development of new materials and new technology. The Moscow No. 12 Power Plant was the best

example of the new presentation of the modern industry [50]. The Moscow No. 12 Power Plant is shown in Figure 18.

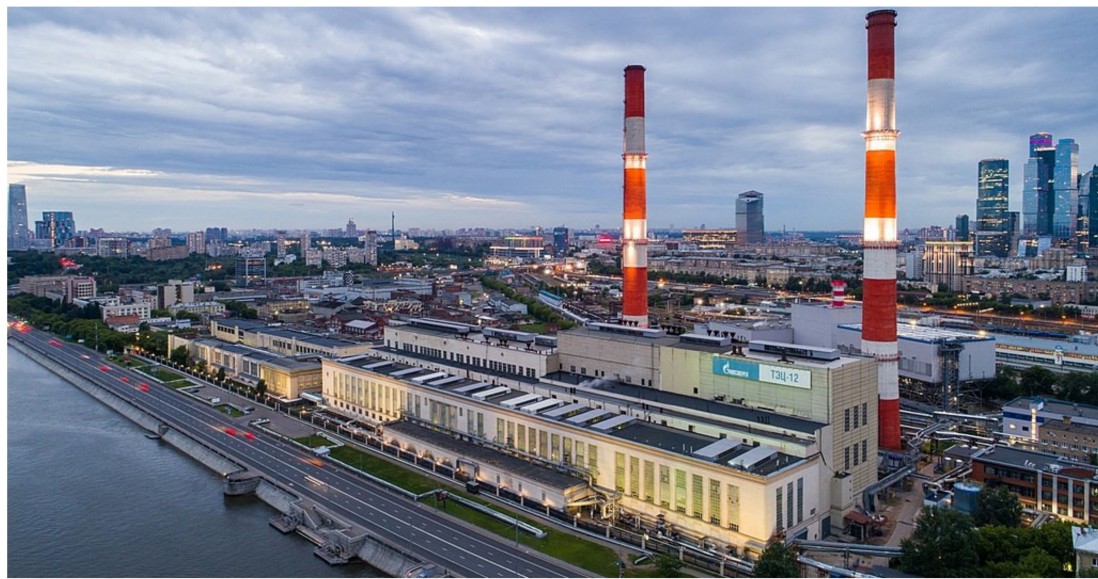

**Figure 18.** Moscow No. 12 Power Plant. (Source: [51]).

### 2.2.7. The Achievements of Modern Industrial Architecture in the Soviet Union

With the review of the development of modern industrial architecture in the Soviet Union from 1917 to 1954, we found that the Soviet Union established its modern industrial architecture, based on constructivism, in the early stage of the design of power plants. In the First Five-Year Plan, the Soviet Union absorbed the achievements of the United States' modern industrial architecture and carried out a lot of exploration and innovation. Modern industrial architecture has been further improved in the Soviet Union, which made its design theory and application method further expanded. The achievements of the Soviet Union are as follows: (1) The Soviet Union absorbed the precast concrete technology from other European countries to improve the standardized design and construction technology; and (2) the Soviet Union furthered the theory of modernism in the industrial architecture and paid more attention to perfect working environment. The garden design became an important content.

### 2.2.8. The Transfer of Modern Industrial Architecture from the Soviet Union to China

China signed the "156 Projects" assistance contract with the Soviet Union in 1950. The Soviet Union began to export large-scale industrial assistance to China. China learned from the Soviet Union in the field of industrialization and carried out China's First Five-Year Plan from 1953 to 1957. In "156 Projects" assistance contract, 150 core industrial projects constituted the foundation and system of China's industry, and more than 900 medium-sized industrial plants were built to serve them [52]. Through the "156 Projects", the Soviet Union exported a large number of technical drawings and industrial equipment to China and sent a lot of experts and consultants in the field of industrial technology and construction engineering to work in China [53].

There were three main channels for the realization of the transfer of modern industrial architecture from the Soviet Union to China. They are as follows: (1) The first channel was training and architectural education. This occurred in both industrial plant construction sites and professional schools. The Soviet Union's scholars helped China establish modern architecture and the standard of structural technology; (2) the second channel was the transfer of technical data. This occurred through various technological standards, industrial plant design drawings, production and manufacturing design

drawings, as well as other technical books. According to statistics, 6536 kinds of technical data were exported to China; and (3) the third channel was taking part in designing tasks. This occurred in the whole process of the planning, design, and construction of the "156 Projects". According to statistics, more than 20,000 architects, engineers, and scholars were sent to China to participate in the design and direction works of the "156 Projects" [54].

*2.3. The Development of Modern Industrial Architecture in China in the 1950s*

2.3.1. The Selection of a Modern Industrial Architecture Importer

At the beginning of the founding of the People's Republic of China, the Chinese government launched the recovery policy (1949–1952) to restore its infrastructure. Transportation restoration received the greatest investment. The railway network in the northeast was given priority. As a result, the railway network became one of the factors influencing the selection of an importer.

Another important factor was the mature import and export trade channels. In fact, the United States had more advantages than the Soviet Union. Since the 1940s, the United States had already been China's largest importer, which meant that China and the United States had established mature and convenient channels for the exchange of goods. The leader, Mao Zedong, had intended to invite the United States to export industrial assistance to China before 1944. He proposed the idea of economic cooperation between China and the United States after World War II, when he met Xie Weisi, a member of the U.S. military observation group. Mao Zedong believed that the two countries would benefit a lot from the process of industrial assistance. China's light industry could provide the United States with a place of investment and an export market for the latter's heavy industrial products. The United States would receive compensation from China's industrial resource materials and agricultural products [55].

It can be observed that the first selection of China's industrial importer was the United States. After World War II, the situation was changed due to the support of the United States for Chiang Kai Shek. As a result, Mao Zedong began to reconsider China's relationship with the United States. China was determined to join an alliance with the Soviet Union [56]. In addition, from the perspective of geo-economics, the Soviet Union borders on China, and there existed a complete railway network in Northeast China, which could be directly connected with the Eurasian railway network of the Soviet Union, and this would facilitate the transportation of equipment, materials, and experts.

In June 1949, Mao Zedong announced, in his article, "Discussion on the People's Democratic Regime", a policy, called the "One-side Down", to join in an alliance with the Soviet Union. In July 1949, the other parties and people's organizations issued a statement to support this policy. In September 1949, this policy was also accepted at the first plenary session of the Chinese People's Political Consultative Conference. In December 1949, Mao Zedong led a delegation to Moscow for an official visit. After nearly two months of negotiations, China signed the assistance contract with the Soviet Union [57].

The Soviet Union eventually became China's selection of the main industrial importer, which led to great changes in China's economic relations and trade direction with the whole world. The transfer of modern industrial architecture from the Soviet Union greatly promoted the development of China's industry. Conversely, it also made China give up the opportunity to learn directly from the United States and other Western countries.

2.3.2. The Transfer Channel of Modern Industrial Architecture from the Soviet Union

Experts were the main transfer channel. In September 1949, K. Abramov, the vice-chairman of the Soviet Union, led a group of experts to Beijing to assist in the research on Beijing's urban planning and construction. He proposed that China's modern architecture should fully present the architectural vision of the national form. In 1952, two architectural experts, L. Muxin and A. Ashepkov, arrived in Beijing. The former served as a senior consultant of the general construction office of the Central Finance and Economics Commission. The latter was hired as a college teacher to teach

industrial architecture at Tsinghua University. These two experts began to systematically introduce the architectural theory of socialist realism and the design methodology of the national form plus the socialist content to China [58]. Due to the vague definition of socialist realism in the Soviet Union, the interpretation of the Soviet Union's experts could not impart to Chinese architects a clear understanding. In the design projects of industrial plants, Chinese architects had to learn from the civil buildings with the socialist realism style in the Soviet Union. As a result, the style of Chinese modern industrial architecture started to deviate from the mainstream trend of the world modern industrial architecture and fell into the quagmire of classical revival, which integrated the Russian Imperial style and Chinese traditional elements.

Another important channel was books and data. In 1951, China received 32,000 technology books and data from the Sciences Academy of the Soviet Union and other official organizations in the Soviet Union. In 1952, the Soviet Union provided nearly 5000 kinds of architecture and construction books. In the same year, China translated and published 756 kinds of planning and design books, 60% of which were industrial architecture books [59]. The transfer of the Soviet Union's technical books and data played an important role in China's absorption of the knowledge of modern industrial architecture and structural technology, which also laid the foundation for the standards and structural engineering of modern architecture in China.

Directing the planning and design of the "156 Projects" was the third channel. The "156 Projects" was the core content of China's First Five-Year Plan, which established a complete industrial system of heavy industry and light industry, which included military industry, energy industry, machinery industry, precision instruments industry, etc. Through cooperation between the Soviet Union's experts and Chinese architects, China's original industrial distribution and construction survey was conducted and finished before 1953. The Chinese government adopted the research report submitted by the Soviet Union's experts and was determined to build an industrial system with three huge industrial base groups, located in the Northeast, Southwest, and Huazhong. The most important construction content was the new industrial plant bases, which included industrial base planning, plant planning, workshop design, civil building design, and so on. Taking Northeast China as an example, the proportion of new industrial base projects was more than 65%. The Soviet Union's experts participated in the whole process the construction and incorporated the Soviet Union's modern industrial architecture and the methodology of the socialist content plus the national form into the projects [60].

### 2.3.3. The Influence of the Chinese Ideological Trend on the Development of Modern Industrial Architecture

In the 1950s, the promotion and interpretation of the Soviet Union's modern industrial architecture were inseparable from the political and cultural environment in China. The ideological trend in architecture was the most important influencing factor. The most famous architect, Liang Sicheng, was the first Chinese architectural expert to learn and introduce the Soviet Union's architectural theory. In December 1952, he issued the article, "The Soviet Union's Experts Helped Correct the Thinking of Architectural Design", in which he summarized socialist realism into four key points: (1) Architecture should be concerned with people's life; (2) architectural art should present a social theme; (3) architecture and urban planning should accompany the social ideology; and (4) architecture should present the national characteristics. In 1953, he spent three months visiting the Soviet Union with the delegation of the Chinese Academy of Sciences. He investigated the development of the Soviet Union's architecture, inspired by the achievements of the Soviet Union's architecture [61]. After returning to China, Liang Sicheng delivered a presentation, entitled "The Study and Application of Socialist Realism and National Heritage in Architectural Art", at the first conference of the Architectural Society of China. He proposed that "it is necessary for architects to learn the theory of socialist realism". He advocated raising the architectural methodology of the national form to the height of support for the government [62]. He also expressed a strong interest in the methodology of the national form and issued the article, "The Characteristics of Chinese Architecture", in 1954, in which he proposed

the feasibility of replacing the composition and components of Western classical architecture with those of Chinese traditional architecture, based on the translatability of classical architecture [63]. Liang Sicheng's approval of the Soviet Union's socialist realism theory, especially the methodology of the nation form, was closely related to his family's cultural background, his growth process, and his study experience. The reasons are as follows: (1) He took over the flag of rejuvenating the nation from his father, Liang Qichao; (2) he paid more attention to architectural history during his study in the University of Pennsylvania in the United States and received the rigorous classical architectural training of the Paris Beaux-Art; (3) he put all his energy into completing the investigation of Chinese traditional architecture in the 1930s; and (4) he supported the Chinese government's policy of joining in an alliance with the Soviet Union in order to be recognized by the Chinese government at the ideological level [64].

Liang Sicheng tried his best to implement the idea of reviving Chinese classicism while serving as the deputy director of the Beijing Urban Planning Commission. Under his strong promotion, the architecture of the Chinese national form prevailed everywhere in China from 1953 to 1954. This influence inevitably spread into the industrial plant of the "156 Projects" [65]. The methodology of the national form became the goal of Chinese architects in exploring modern architecture, which had a negative impact on the development of modern industrial architecture in China. It made industrial plants apply the modern planning concept and new structural technology while presenting the facade in the classical revival style.

## 3. Methods

Based on the previous literature review, we described the global transfer of modern industrial architecture from the United States to the Soviet Union and then to China. For the purpose of further proving the relevance of the modern industrial architecture among the three countries and finding the inheritance and changes existing in the "156 Projects", we were determined to conduct field investigation immediately. In this process, we also implemented careful measurement and face-to-face interviews to understand the status of the workshops and achieve the objective assessment of their multi-value. The results would not only serve the evidence to prove the origin of China's modern industrial architecture but also benefit the conservation and regeneration of them in the future.

### 3.1. Field Investigation

### 3.1.1. The Inheritance of Modern Industrial Architecture in China

Jilin province, Heilongjiang province, and Sichuan province are the most important industrial provinces in China in the 1950s. A lot of industrial plants were built in the capital cities of these provinces. Three core plants of "156 Projects", the Changchun First Automobile Works, the Harbin Boiler Factory, and the Chengdu Hongguang Electronic Transistor Factory were selected as the study cases since their distinctive features. They all represented the outstanding achievement of China's modern industrial architecture, which helped us easily observe the inheritance and compare the changes among the United States, the Soviet Union, and China from the perspective of the planning concept, design theory, and structural technology. In addition, the industrial heritages in these cases are all facing the difficulties of conservation and regeneration. A deep field investigation is urgent and necessary. Field investigation was conducted in the first half of 2019. We investigated 13 workshops in the three cases. The details of the samples are listed in Table 1.

**Table 1.** The details of the 13 samples.

| Location | Case | Workshop | Structure | Built Year |
|---|---|---|---|---|
| Changchun, Jilin Province | Changchun First Automobile Works | Punching Press Workshop | Steel frame | 1954 |
| | | Cutting Workshop | concrete frame + Steel truss | 1954 |
| | | Foundry Workshop | concrete frame + Steel truss | 1954 |
| | | Welding Workshop | Steel frame | 1954 |
| | | Assembly Workshop | concrete frame + Steel truss | 1955 |
| | | Power Station | concrete frame | 1953 |
| Harbin, Heilongjiang Province | Harbin Boiler Factory | Cutting Workshop | Steel frame | 1955 |
| | | Foundry Workshop | concrete frame + Steel truss | 1955 |
| | | Welding Workshop | Steel frame | 1956 |
| | | Assembly Workshop | concrete frame + Steel truss | 1956 |
| Chengdu, Sichuan Province | Chengdu Hongguang Electronic Transistor Factory | Office Building | concrete frame | 1957 |
| | | Testing Workshop | Steel frame | 1957 |
| | | Assembly Workshop | concrete frame + Steel truss | 1957 |

The Changchun First Automobile Works was the epitome of China's achievements in modern industrial architecture. It not only inherited the achievements of the United States' plant planning and structural technology, but also reflected the influence of the Soviet Union's socialist realism theory. It was designed by the Institute of Automobile and Tractor Industrial Design in Moscow. The total construction area was 702,480 m$^2$, the workshop area was 382,274 m$^2$, and the residence area was 320,206 m$^2$. The total investment was 650 million yuan. The construction started in July 1953 and was completed in July 1956 [66]. The Changchun First Automobile Works applied the complete planning of a huge industrial base, which contained the main plant planning, the supporting area planning, the living area planning, and future development area planning. It drew a lot from the Gorkovsky Avtomobilny Zavod in the Soviet Union and also presented the application of the successful experience of the Ford Rouge Complex in the United States [67]. The planning of the industrial base is shown in Figure 19.

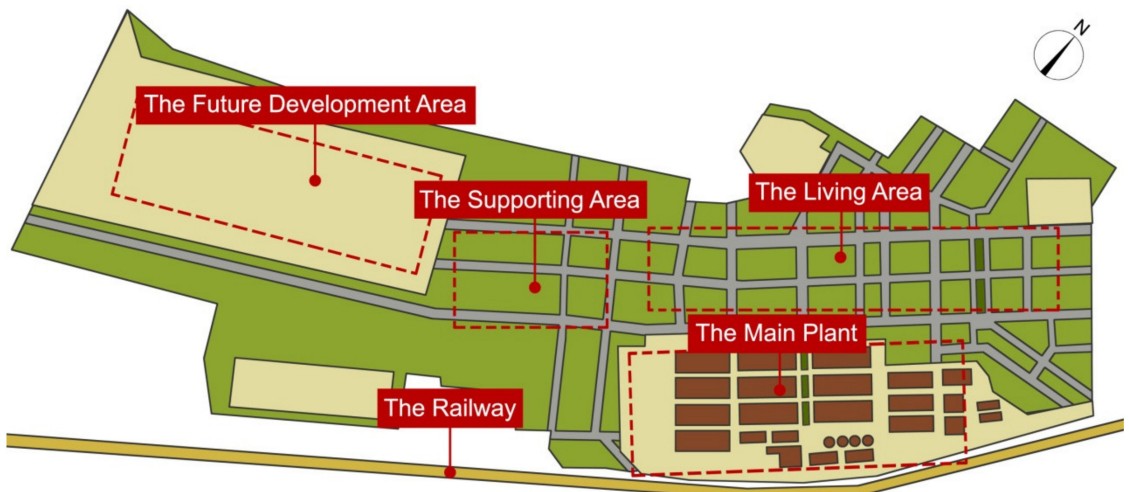

**Figure 19.** The industrial base general planning of the Changchun First Automobile Works.

There was a clear axis and lucid logic in the planning of the main plant of the Changchun First Automobile Works. The planning absorbed and applied the methodology of the socialist content, which expressed a strong centripetal order and humanistic care for workers. The main plant was divided into the east workshops zone and the west workshops zone by the central-axis green garden [68]. All the workshops in the two zones were arranged in a centralized way and designed into huge workshops, according to the needs of the assembly line. The power plant and material transportation were close to the railway at the southeast of the plant. The future development area was reserved in the southwest of the industrial base. The traffic flow of the plant was organized reasonably, and a separation of the people flow and vehicle flow was implemented. The green garden effectively regulated the ecological environment in the plant and created a pleasant and comfortable working environment for workers. The main plant of the Changchun First Automobile Works is shown in Figure 20, and outside environment in the main plant is shown in Figure 21.

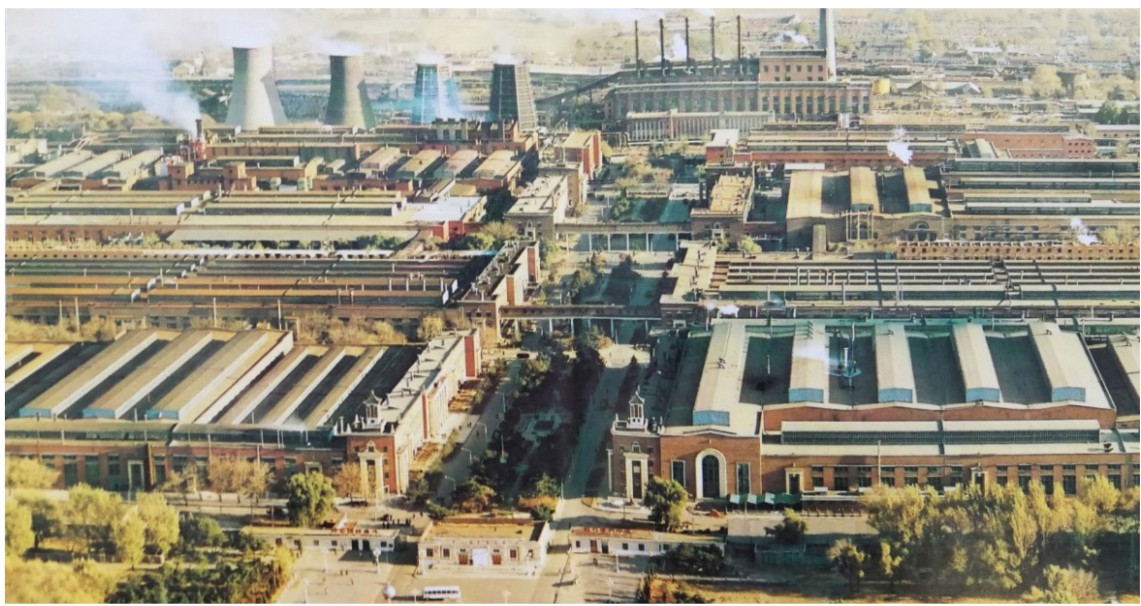

**Figure 20.** The main plant of the Changchun First Automobile Works. (Source: [66]).

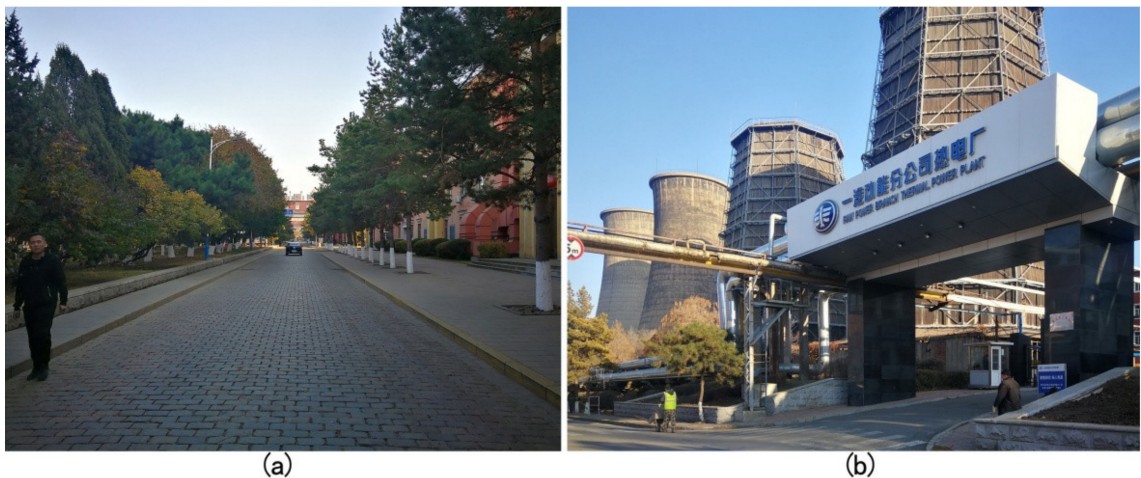

**Figure 21.** The outside environment in the main plant: (**a**) the central-axis green garden along the main road and (**b**) the entrance of Power Station.

The structure of the workshops all adopted the Soviet Union's standardized design. The size of the column net was 6 × 18 m. The structural system consisted of reinforced concrete columns and a

steel frame roof truss. There were roof windows in a northeast–southwest orientation, which could be opened manually for internal ventilation. The envelope walls were made of red bricks with a double-layer [69]. The facade fully presented the Chinese national form. It took the Soviet Union's classicism as the keynote and paid attention to the decoration details of the outside column, cornice, and window cover line. At the same time, it organically integrated traditional Chinese architectural symbols and elements. The architects designed two Chinese-style pagodas that had four-corner cusp roofs on the top of the workshops near the entrance of the plant. A lot of traditional Chinese patterns emerged on the walls between windows as decorations. This innovative design formed a collision and blending of Chinese and Western architectural culture, which enriched the connotation of the Chinese national form. Chinese modern industrial architecture was gradually created and became a unique and fascinating sample of the world's modern industrial architectural history. The facade of Punching Press Workshop and Cutting Workshop are shown in Figure 22, and their structure are shown in Figure 23.

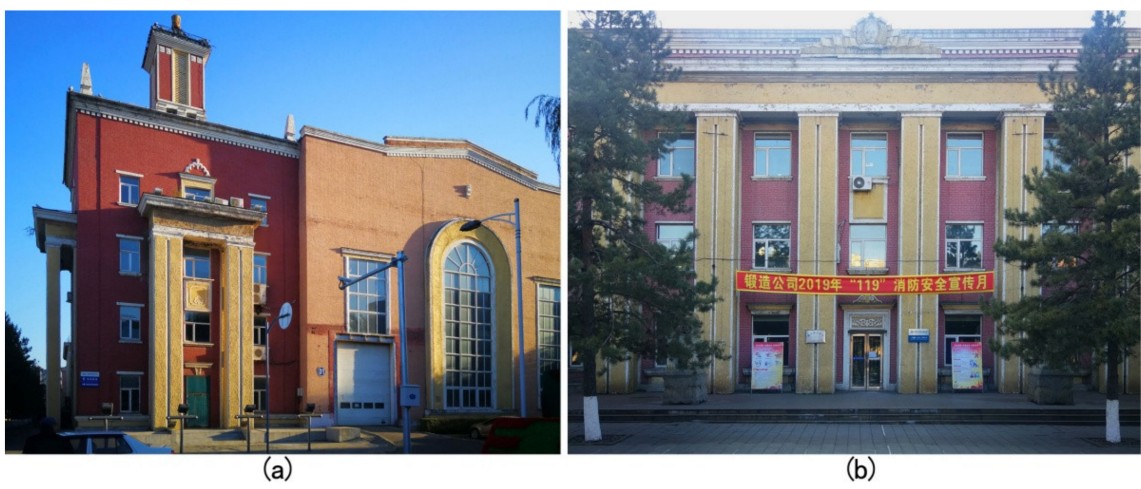

**Figure 22.** The facade of the workshops in the Changchun First Automobile Works: (**a**) Punching Press Workshop and (**b**) Cutting Workshop.

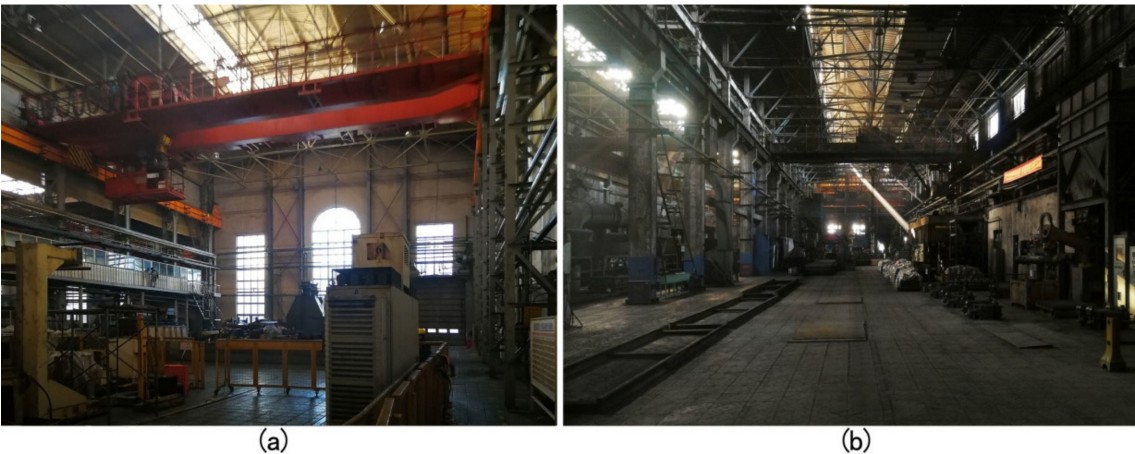

**Figure 23.** The structure of the workshops in the Changchun First Automobile Works: (**a**) Punching Press Workshop and (**b**) Cutting Workshop.

### 3.1.2. The Changes of Modern Industrial Architecture in China

The Changchun First Automobile Works presented the continuity and creativity in the global transfer process of modern industrial architecture. As the first batch of core plant bases of the "156 Projects", the Changchun First Automobile Works fully presented the inheritance of the Soviet Union's modern industrial architecture and showed the innovation based on Chinese ideological trend.

However, this situation was fleeting, and some minor changes gradually happened. Due to the high cost of construction and the implementation of the Anti-waste Movement at the end of the 1950s in China, the new architecture that expressed the Chinese national form was seriously suppressed. As a result, the second batch of core industrial bases, built from 1955 to 1958, were inevitably and constantly simplified, and began to display a clear modernist style [70]. The Harbin Boiler Factory, built in 1955, had begun to change its design ideas in the plant planning and workshop design. The architects had simplified the unnecessary decoration and only designed a small amount of cornicing on the facade of the workshop. They also compressed the area of the garden in the plant in order to expand the construction area for the workshops. In the third case, the Chengdu Hongguang Electronic Transistor Factory, which started construction in 1957, had been designed in the pure modernist style. All the decoration on the facade was canceled. The above process indicated that Chinese modern industrial architecture had begun to display a stronger and clearer trend of modernism. After a short attempt at interpreting the Chinese national form, the Changchun First Automobile Works became an unique case, with a positive innovation and no successor in the development of modern industrial architecture in China. The roof and facade design of the workshops in the Harbin Boiler Factory are shown in Figure 24, their structure is shown in Figure 25. The facade design of the workshops in the Chengdu Hongguang Electronic Transistor Factory is shown in Figure 26, their structure is shown in Figure 27.

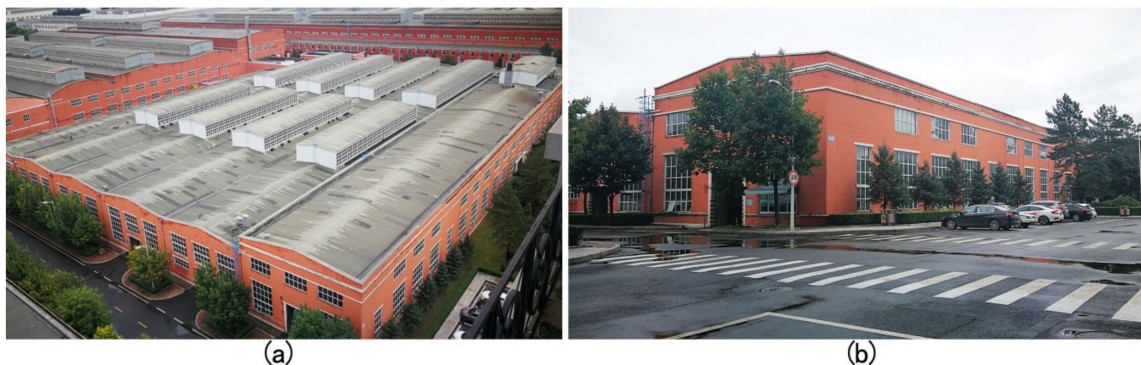
(a)                                                                                                                (b)

**Figure 24.** The Harbin Boiler Factory: (**a**) the roof and the external design of Welding Workshop and (**b**) the facade of Welding Workshop.

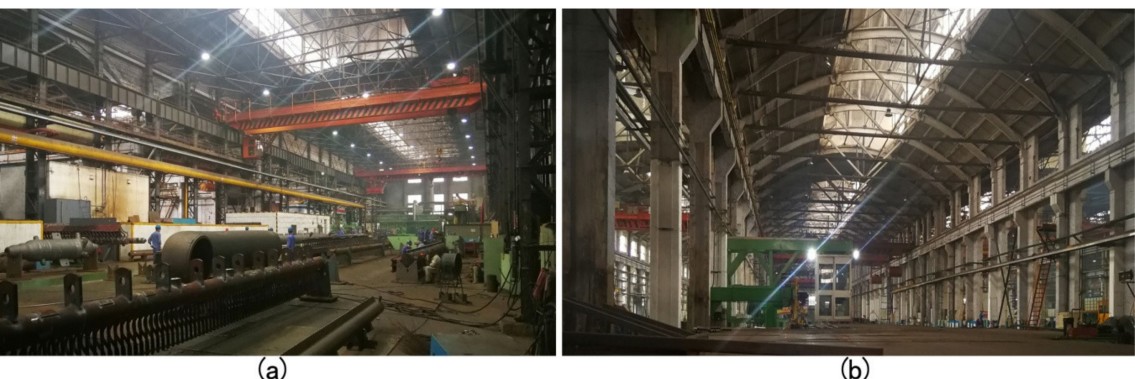
(a)                                                                                                                (b)

**Figure 25.** The Harbin Boiler Factory: the structure of (**a**) Welding Workshop and (**b**) Foundry Workshop.

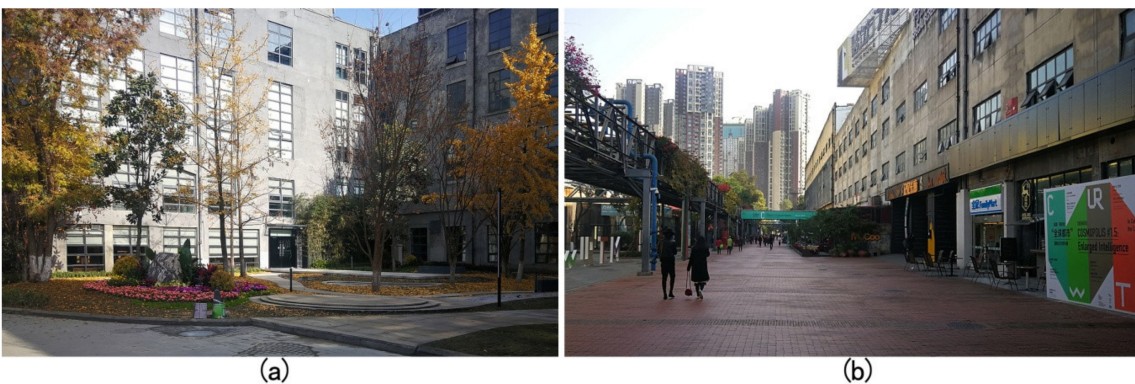

**Figure 26.** The Chengdu Hongguang Electronic Transistor Factory: the facade of (**a**) Office Building and (**b**) Assembly Workshop.

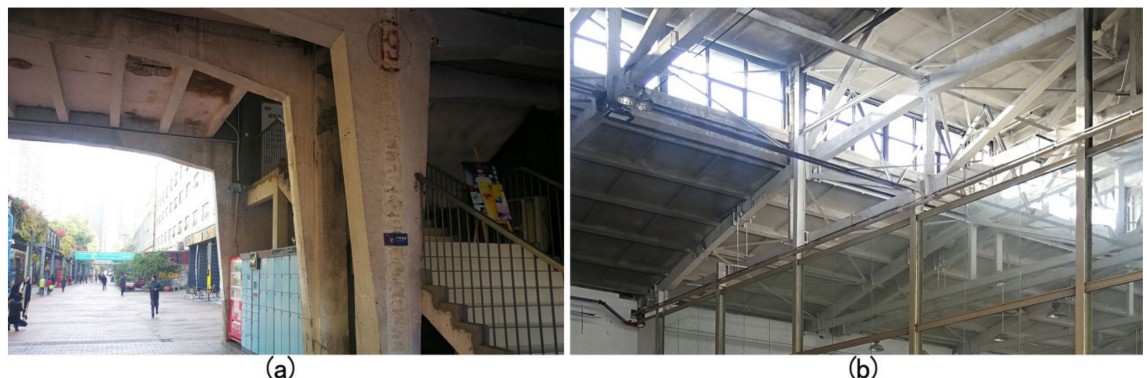

**Figure 27.** The Chengdu Hongguang Electronic Transistor Factory: (**a**) the reinforced concrete structure of Assembly Workshop (**b**) the steel frame roof truss of Testing Workshop.

*3.2. Measurement: The Integrity and Damage of the Modern Industrial Heritage*

In the second half of 2019, we began to measure the integrity and damage of the workshops. The roof and facade was scanned with DJL Mavic-2 Zoom instrument (DJL Corp., China), and the deformation and shift of the structure was measured with SNDWAY SW-120A instrument (SNDWAY Corp., China). The process of the measurement and the typical damage details are shown in Figures 28 and 29, respectively.

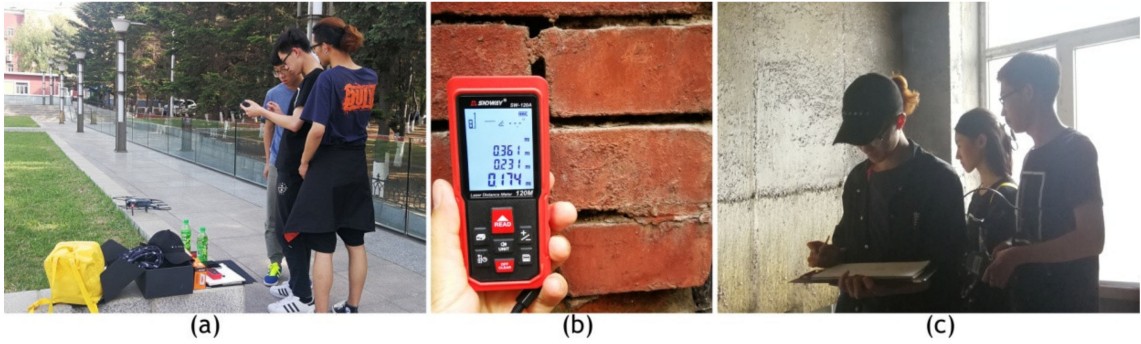

**Figure 28.** The process of the measurement: (**a**) scanning the roof and facade, (**b**) measuring the envelope, and (**c**) recording the damage detail.

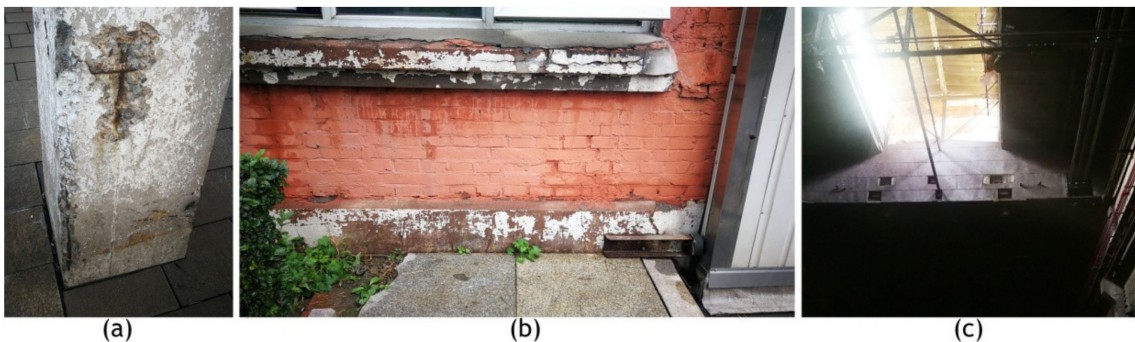

**Figure 29.** The typical damage details of the workshops: (**a**) the structural damage of Foundry Workshop in the Changchun First Automobile Works, (**b**) the facade damage of Welding Workshop in the Harbin Boiler Factory, and (**c**) the roof damage of Cutting Workshop in the Changchun First Automobile Works.

We established a statistical table of integrity and damage based on the record of the measurement. The integrity problems mainly included facade integrity, roof integrity, and component integrity. We measured all the 13 samples in three cases, recorded 608 facade integrity problems, 113 roof integrity problems, and 602 building component integrity problems. The damage problems referred to the structure damage. We recorded 156 structure damage problems, 44 of which were serious problems, mainly emerged on reinforced concrete columns. According to the statistics, we established a general assessment for future conservation and regeneration. There are three levels in the general assessment as follows: (1) Worse level with 3 scores: more than 140 integrity and damage problems; (2) normal level with 2 scores: 90–140 integrity and damage problems; and (3) better level with 1 score: less than 90 integrity and damage problems. The statistics of integrity and damage is listed in Table 2, and the general assessment is listed in Table 3.

**Table 2.** The statistics of integrity and damage of the 13 samples.

| Case | Workshop | Integrity Problem | | | Structure Damage | |
|---|---|---|---|---|---|---|
| | | Facade | Roof | Component | Normal | Serious |
| Changchun First Automobile Works | Punching Press Workshop | 48 | 6 | 37 | 9 | 1 |
| | Cutting Workshop | 38 | 11 | 32 | 11 | 2 |
| | Foundry Workshop | 68 | 14 | 77 | 16 | 9 |
| | Welding Workshop | 59 | 7 | 66 | 15 | 11 |
| | Assembly Workshop | 70 | 19 | 80 | 9 | 7 |
| | Power Station | 45 | 10 | 35 | 12 | 2 |
| Harbin Boiler Factory | Cutting Workshop | 42 | 9 | 30 | 4 | 2 |
| | Foundry Workshop | 55 | 10 | 49 | 2 | 0 |
| | Welding Workshop | 43 | 3 | 31 | 6 | 2 |
| | Assembly Workshop | 51 | 7 | 72 | 8 | 1 |
| Chengdu Hongguang Electronic Transistor Factory | Office Building | 22 | 5 | 25 | 5 | 2 |
| | Testing Workshop | 27 | 5 | 29 | 9 | 1 |
| | Assembly Workshop | 40 | 7 | 39 | 6 | 4 |

**Table 3.** The general assessment of the 13 samples.

| Case | Workshop | General Assessment | | |
| --- | --- | --- | --- | --- |
| | | Worse (3) | Normal (2) | Better (1) |
| Changchun First Automobile Works | Punching Press Workshop | | √ | |
| | Cutting Workshop | | √ | |
| | Foundry Workshop | √ | | |
| | Welding Workshop | √ | | |
| | Assembly Workshop | √ | | |
| | Power Station | | √ | |
| Harbin Boiler Factory | Cutting Workshop | | | √ |
| | Foundry Workshop | | √ | |
| | Welding Workshop | | | √ |
| | Assembly Workshop | | √ | |
| Chengdu Hongguang Electronic Transistor Factory | Office Building | | | √ |
| | Testing Workshop | | | √ |
| | Assembly Workshop | | √ | |

*3.3. Face-to-Face Interview: The Multi-Value of the Modern Industrial Heritage*

We conducted face-to-face interviews for multi-value such as historical, economic, aesthetic, and technological value at the end of 2019. 137 people working in the Changchun First Automobile Works, 115 people working in the Harbin Boiler Factory, and 122 people working in the Chengdu Hongguang Electronic Transistor Factory, were surveyed. About 35% of them were 45–55 years old and employed more than 25 years; 30% of them were 35–44 years old, and employed more than 15 years; 35% of them were 20–34 years old, and employed more than 10 years. They all witnessed the historical changes of these three plants in different eras, and presented relatively objective assessment according to their different ages, world outlook, and aesthetic taste. The interview analysis of the historical and economic value is shown in Figure 30. The interview records of aesthetic and technological value are shown in Figure 31.

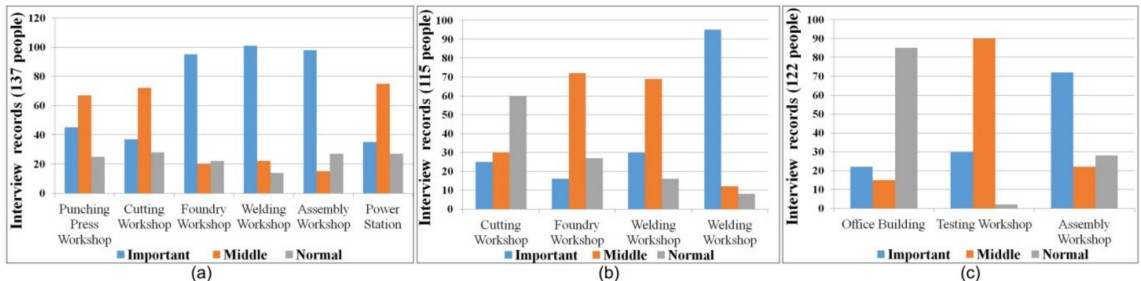

**Figure 30.** The interview analysis of historical and economic value: (**a**) the Changchun First Automobile Works, (**b**) the Harbin Boiler Factory, and (**c**) the Chengdu Hongguang Electronic Transistor Factory.

According to the statistics of the interview records, we established the assessment of historical and economic value and aesthetic and technological value. Different scores (1–3) present different value level objectively, which provides us a reference for conservation and regeneration. The assessment of historical and economic value is listed in Table 4. The assessment of aesthetic and technological value is listed in Table 5.

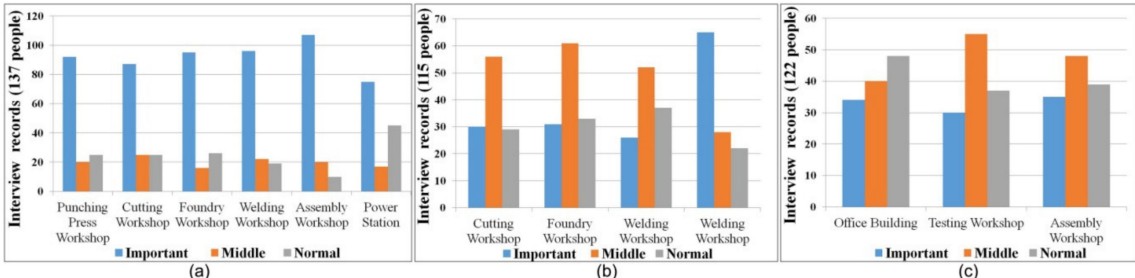

**Figure 31.** The interview analysis of aesthetic and technological value: (**a**) the Changchun First Automobile Works, (**b**) the Harbin Boiler Factory, and (**c**) the Chengdu Hongguang Electronic Transistor Factory.

**Table 4.** The assessment of historical and economic value of the 13 samples.

| Case | Workshop | Historical and Economic Value | | |
| --- | --- | --- | --- | --- |
| | | Important (3) | Middle (2) | Normal (1) |
| Changchun First Automobile Works | Punching Press Workshop | | √ | |
| | Cutting Workshop | | √ | |
| | Foundry Workshop | √ | | |
| | Welding Workshop | √ | | |
| | Assembly Workshop | √ | | |
| | Power Station | | √ | |
| Harbin Boiler Factory | Cutting Workshop | | | √ |
| | Foundry Workshop | | √ | |
| | Welding Workshop | | √ | |
| | Assembly Workshop | √ | | |
| Chengdu Hongguang Electronic Transistor Factory | Office Building | | | √ |
| | Testing Workshop | | √ | |
| | Assembly Workshop | √ | | |

**Table 5.** The assessment of aesthetic and technological value of the 13 samples.

| Case | Workshop | Aesthetic and Technological Value | | |
| --- | --- | --- | --- | --- |
| | | Important (3) | Middle (2) | Normal (1) |
| Changchun First Automobile Works | Punching Press Workshop | √ | | |
| | Cutting Workshop | √ | | |
| | Foundry Workshop | √ | | |
| | Welding Workshop | √ | | |
| | Assembly Workshop | √ | | |
| | Power Station | √ | | |
| Harbin Boiler Factory | Cutting Workshop | | √ | |
| | Foundry Workshop | | √ | |
| | Welding Workshop | | √ | |
| | Assembly Workshop | √ | | |
| Chengdu Hongguang Electronic Transistor Factory | Office Building | | | √ |
| | Testing Workshop | | √ | |
| | Assembly Workshop | | √ | |

*3.4. Results*

Due to facing the dilemma of conservation and regeneration, more than 80% of the workshops have become industrial heritage. A large number of integrity and damage problems were appearing, 41.2% of which emerged on the facade, 7.6% of which emerged on the roof, 40.7% of which emerged on the components, and 10.5% of which emerged on the structure. The statistics revealed that these industrial heritages have good structural stability, which proved the necessity of conservation and the possibility of regeneration. Based on a face-to-face interview, we have confirmed that nearly 40% of them have a high historical and economic value and nearly 60% of them have a high aesthetic and technological value. As a result, how to better implement conservation and regeneration became the key.

## 4. Discussion

China totally absorbed the Soviet Union's planning concept of a huge industrial base and individual plant, which caused the formal sense to have more priority than the functional demands in determining the planning. In order to meet the needs of a longer assembly line, the huge workshops were built with a column net of $6 \times 18$ m, which made it impossible to reserve a reasonable internal space for industrial upgrading right now. After almost 70 years, there are a large number of integrity problems on the facade, the roof, and the component, as well as the damage problems on the structure. The majority of the workshops in three cases became the industrial heritage since losing their manufacturing function. In order to conserve the unique sample of the industrial civilization and re-stimulate the urban economy, developing sustainable modern industrial tourism attracts our research interest.

The Outline of China's Industrial Tourism Development (2016) released that 1000 national industrial tourism demonstration projects and 10 industrial tourism cities will be established in China until 2025. Changchun, Harbin, and Chengdu were all listed in the catalog. China will enter a golden development period of industrial tourism from 2020 to 2025. The total direct income will exceed 200 billion yuan, the number of new direct tourism employment will exceed 1.2 million, and the total comprehensive income for the regions and cities may exceed 10 times the direct income [71]. Through the statistics of bibliometrics, up to the end of 2019, more than 70 regions and cities in China have carried out relate studies on the development of industrial tourism projects, but the successful cases are still in lack because of the shortage of sustainable strategies [72]. We proposed two sustainable development strategies of industrial tourism based on the comprehensive assessment.

Firstly, we calculated and counted the assessment scores for each workshop heritage based on the above assessments Tables 3–5. We suggested reasonable reinforcing and repairing should be implemented according to the comprehensive assessment results. We recommend the following: (1) The workshops with high assessed score (more than 7) should be reinforced immediately, especially on their structure. We could increase the section area of the concrete columns and replace the broken components of the steel structure; (2) the workshops with middle assessed score (5–7) should be repaired on both facade and roof. We must respect their original aesthetic appearance and use original construction methods, building materials, and decoration color; and (3) the workshops with low assessed score (less than 5) should be minorly patched on parts of the walls, windows, stairs, and other components. The above three measures will serve the sustainability of architectural material space. In addition, we proposed that preserving the mechanical equipment and exhibiting the original manufacturing process would be necessary and beneficial for serving the sustainability of industrial culture. The industrial activities and workers' spiritual experience were the main content of the industrial culture. They were all spatialized by the equipment and assembly line in the architectural space. A successful and remarkable industrial tourism project should make tourists easily understand the manufacturing process and better enjoy the complicated mental feeling of self-identify brought by the industrial activities, collective solidarity stimulated by the cooperation, and individual alienation caused by the division of assembly line. The comprehensive assessment and suggestions are listed in Table 6.

**Table 6.** The comprehensive assessment and suggestions of the 13 samples.

| Case | Workshop | Comprehensive Assessment | Suggestion |
|---|---|---|---|
| Changchun First Automobile Works | Punching Press Workshop | 7 | Repairing the facade |
| | Cutting Workshop | 7 | Repairing the facade |
| | Foundry Workshop | 9 | Reinforcing the structure |
| | Welding Workshop | 9 | Reinforcing the structure |
| | Assembly Workshop | 9 | Reinforcing the structure |
| | Power Station | 7 | Repairing the facade |
| Harbin Boiler Factory | Cutting Workshop | 4 | Minor patching |
| | Foundry Workshop | 6 | Repairing the facade |
| | Welding Workshop | 5 | Repairing the facade |
| | Assembly Workshop | 8 | Reinforcing the structure |
| Chengdu Hongguang Electronic Transistor Factory | Office Building | 3 | Minor patching |
| | Testing Workshop | 5 | Repairing the facade |
| | Assembly Workshop | 7 | Repairing the facade |

Secondly, we established achievable and long-term goals and presented two reasonable patterns for the sustainable development of China's modern industrial tourism. We recommend the following: (1) The goal of informational spillover effect: the database of industrial heritage resources should be established, including the basic information, the GIS data, and assessment results of multi-value, which can be used as a reference for future conservation, regeneration and tourism development; (2) the goal of environmental and functional spillover effect: the industrial tourism projects should extend ecological regulation capacity and provide more service content to improve their sustainability. On the one hand, we could expand the green garden areas and purify the water circulation system in the plants; on the other hand, we suggest adding the functions of technology exhibition, culture education, and leisure vacation into the above cases [73]; and (3) the goal of regional economic spillover effect: the comprehensive industrial tourism projects with special themes should be developed according to their unique characters and advantages. The Changchun First Automobile Works and the Harbin Boiler Factory are located on the important node cities, Changchun and Harbin, which are linked by the high-speed railway network. Both of the plants presented the achievement of China's heavy industry and exploration of modern industrial architecture. Accordingly, we could develop and design an urban linkage regional heavy industrial tourism project (Belt Pattern) based on the convenient railway, consisting of several supporting facilities located along the railway between the two cities. The Chengdu Hongguang Electronic Transistor Factory is located in the capital city of Sichuan Province, the center of China's electronic industry. We could develop and design a centralized electronic industry tourism project (Radiation Pattern), consisting of high technology enterprises, scientific research institutes, and outstanding universities around Chengdu, serving and supporting the main project. The above two patterns will stimulate the rapid development of regional economy and integration of advantageous industrial resources [74–76]. The patterns of urban linkage regional heavy industrial tourism project and the centralized electronic industry tourism project are shown in Figures 32 and 33, respectively.

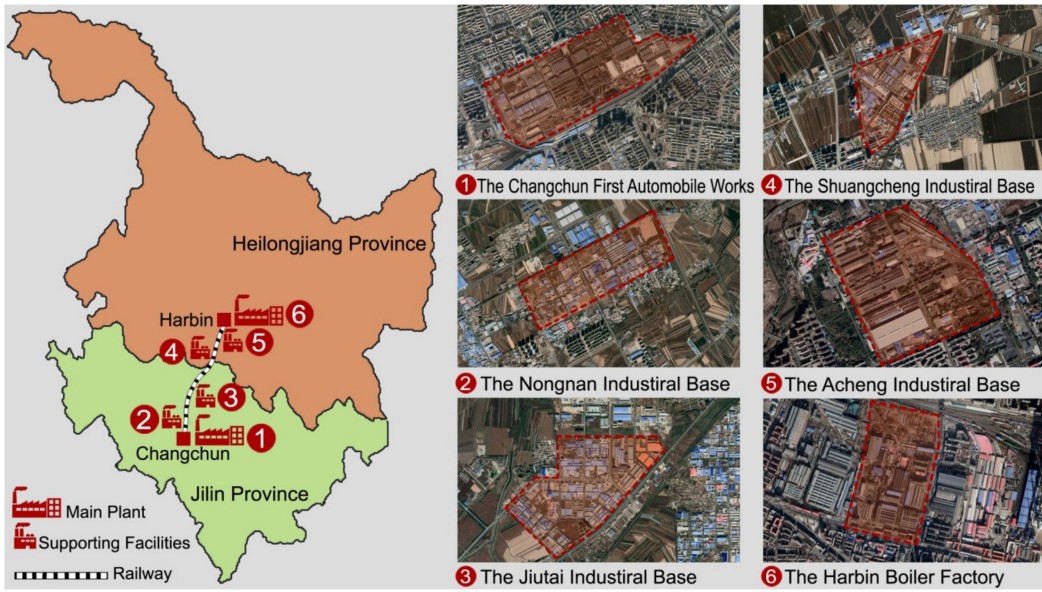

**Figure 32.** The urban linkage regional heavy industrial tourism project (Belt Pattern).

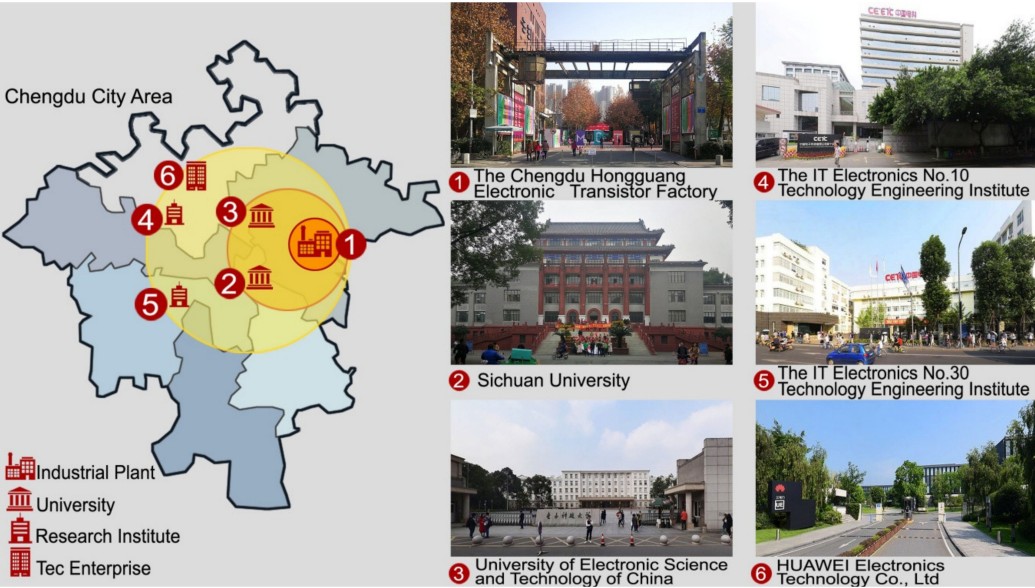

**Figure 33.** The centralized electronic industry tourism project (Radiation Pattern).

## 5. Conclusions

There existed a clear route of the global transfer of modern industrial architecture from the United States to the Soviet Union and then to China in the first half of the 20th century. The origin of China's modern industrial architecture came from the achievement of the United States' modern industrial architecture, meanwhile adding the Soviet Union's standard construction technology. Both of them were integrated into a new industrial architectural vision by the prevailing ideology in China in the 1950s.

Modern industrial architecture, founded and perfected by the American architects, was imported into the Soviet Union during its First Five-Year Plan (1928–1932) through the channel of directing projects and training architects. There was no phenomenon of exclusion in the process of the transfer of modern industrial architecture from the United States to the Soviet Union. In China's First Five-Year Plan (1953–1957), the Soviet Union's modern industrial architecture was exported to China through the

assistance of the "156 Projects". The transfer channel included sending experts, directing projects, and interpreting books. In order to meet the needs of the Chinese domestic political culture and express the prevailing ideological trend of architectural design, a unique style of the modern industrial architecture was invented. The facade vision of the workshops expressed the Chinese national form, which became a precious and valuable sample of the development process of human industrial civilization. At the end of the 1950s, the decoration of China's modern industrial architecture was simplified and began to display a clearer modernist style which was more closed to the United States.

China's modern industrial architecture have high multi-value. Besides, the problems of integrity and damage were extremely obvious. We found that 1323 integrity problems (89.5% of the total) emerged on the facade, the roof, and the component since the aging of building materials, and 156 damage problems (10.5% of the total) emerged on the reinforced concrete columns since the concrete damage. The majority of the workshops became the industrial heritages, which are facing the dilemma of conservation and regeneration. Based on the comprehensive assessment, we proposed two sustainable development strategies and two patterns for the industrial tourism projects as following:

- In order to improve the sustainability of both architectural material space and industrial culture, we should classify the industrial heritages and implement different measures of repairing and reinforcing, meanwhile preserving the mechanical equipment and exhibiting the original manufacturing process.
- A database of industrial heritage resources should be established for future reference and management. Providing more service content and extending ecological regulation capacity could realize the goal of functional and environmental spillover effect. We presented "Belt Pattern" and "Radiation Pattern" for the tourism development of the cases, which depend on their different themes and respective advantages.

**Author Contributions:** Conceptualization: R.H.; Investigation: R.H. and D.L.; Methodology: D.L. and P.C.; Writing original draft: R.H.; Review and editing: D.L. and P.C. All authors have read and agreed to the published version of the manuscript.

**Funding:** This research was funded by the National Art Fund (2019-A-05-(373)-1080); Collaborative Education Project of Ministry of Education (201801154018); Jilin Province Social Science Fund (2018B172); Jilin Educational Science Planning Project (GH180378); China Scholarship Council (2019-44:201905975005).

**Acknowledgments:** We would like to thank the anonymous reviewers for their constructive and supportive feedback.

**Conflicts of Interest:** The authors declare no conflict of interest.

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
