# Peer review of "A Study on the Origin of China’s Modern Industrial Architecture and Its Development Strategies of Industrial Tourism"

_sustainability, doi:10.3390/su12093609_

Round 1

Reviewer 1 Report

A stimulating and comprehensive paper with a coherent narrative discussing the transfer of industrial architecture between Europe, US, Soviet Union and China taking the architecture of Albert Khan as a paradigm. I have little sustained critique as I think this is of publishable standard as it stands. However I have a couple of points.

The paper is mostly strong until Section 5, which reads differently in relation to the rest of the paper. The methodology changes and the tone is more technical rather than historical-reflective. Perhaps this is due to its framing for a special issue. It might be worthwhile revisiting this section as it lacks the lucidity of elsewhere. 

I wonder if there is more to be said on the representational aspect of industrial architecture and its framing as a place of labour relations. There is a paradoxical ethos: partly one of collective solidarity and at the same time individual alienation. If the factory was the place where that spirit was formerly spatialised, what has become of the significance of that ethos in today's industrial architecture? Is that ethos signified in any way in what the authors call "industrial tourism" today? What has happened to the social or cultural sustainability of that ethos today?

Good images but some are low resolution. Figs 2, 9, 10, 12, 19, 31 all needing a resolution check. 

Formatting of references unusual. I can't seem to work this out, which is neither a bibliography nor a footnote. Please clarify with journal style guide for consistency.

Author Response

Dear reviewer,

Thank you for handling the review of our manuscript entitled “A Study on the Origin of China's Modern Industrial Architecture and Its Development Strategies of Industrial Tourism (Sustainability-775648)”. In this revised version, we have carefully addressed all the issues raised by you. We sincerely appreciate your insightful comments and suggestions that greatly help improve our manuscript.

The following is a summary of the point-to-point response and revisions we have made to each of your comments. We look forward to hearing your more suggestion and the outcome of this latest version of the manuscript.

Point 1: A stimulating and comprehensive paper with a coherent narrative discussing  the transfer of industrial architecture between Europe, US, Soviet Union and China taking the architecture of Albert Khan as a paradigm. I have little sustained critique as I think this is of publishable standard as it stands. However I have a couple of points. The paper is mostly strong until Section 5, which reads differently in relation to the rest of the paper. The methodology changes and the tone is more technical rather than historical-reflective. Perhaps this is due to its framing for a special issue. It might be worthwhile revisiting this section as it lacks the lucidity of elsewhere.

Response 1: Thank you for your suggestion which benefits us a lot. We have made a moderate revision for the structure of the manuscript, and tried to make the methodology, discussion, and conclusion clearer. The new version is as follows: 1. Introduction; 2. Background (we described the development of the modern industrial architecture in the United States (section 2.1), the Soviet Union (section 2.3), and China (section 2.4).); 3. Methods (field investigation: the inheritance and changes of modern industrial architecture in China (section 3.1), measurement: the integrity and damage of the modern industrial heritage (section 3.2), and face-to-face interview: the multi-value of the modern industrial heritage (section 3.3).); 4. Discussion (we clarified the origin of China's modern industrial architecture, analyzed the dilemma of conservation and regeneration, and proposed sustainable development strategies for the industrial heritage); and 5. Conclusion.

Firstly, in order to enhance the technicality of the manuscript and clarify the methodology, we divided the origin Section 4 and origin Section 5 into different parts, and integrated the contents of the field investigation, the measurement, the face-to-face interview, the statistics of integrity and damage, and the multi-value assessment together into the new section: 3. Methods. We have revised the following paragraph (Line 632-640, Line 753-762, and Line 767-776), and added new Table 2 (Line 764), new Figure 30, 31 (Line 777-784) in new 3. Methods for further discussion.

(Line 632-640) "Jilin province, Heilongjiang province, and Sichuan province are the most important industrial provinces in China in the 1950s. A lot of core industrial plants were built in the capital cities of these provinces. The Changchun First Automobile Works, the Harbin Boiler Factory, and the Chengdu Hongguang Electronic Transistor Factory were selected as the study cases since their distinctive features. They all represented the outstanding achievement of China's modern industrial architecture, which helps us easily observe the inheritance and compare the changes among the United States, the Soviet Union, and China from the perspective of the planning concept, design theory, and structural technology. In addition, the industrial heritages in these cases are all facing the difficulties of conservation and regeneration. A deep field investigation is urgent and necessary."

(Line 753-762) "We established a statistical table of integrity and damage based on the record of the measurement. The integrity problems mainly included facade integrity, roof integrity, and component integrity. We measured all the 13 samples in three cases, recorded 608 facade integrity problems, 113 roof integrity problems, and 602 building component integrity problems. The damage problems referred to the structure damage. We recorded 156 structure damage problems, 44 of which were serious problems, mainly emerged on reinforced concrete columns. According to the statistics, we established a general assessment for future conservation and regeneration. There are three levels in the general assessment as follows: (1) Worse level with 3 scores: more than 140 integrity and damage problems; (2) normal level with 2 scores: 90-140 integrity and damage problems; and (3) better level with 1 score: less than 90 integrity and damage problems."

(Line 767-774) "We conducted face-to-face interview for multi-value such as historical, economic, aesthetic, and technological value at the end of 2019. 137 people working in the Changchun First Automobile Works, 115 people working in the Harbin Boiler Factory, and 122 people working in the Chengdu Hongguang Electronic Transistor Factory, were surveyed. About 35% of them were 45-55 years old and employed more than 25 years; 30% of them were 35-44 years old, and employed more than 15 years; 35% of them were 20-34 years old, and employed more than 10 years. They all witnessed the historical changes of these three plants in different eras, and presented relatively objective assessment according to their different ages, world outlook, and aesthetic taste."

Secondly, in order to clarify the background of the development of industrial tourism and improve the lucidity of the discussion, we have revised the following paragraph (Line 793-802 and Line 834-857), and added the following paragraph (Line 803-813) in new 4. Discussion (Page 28-30).

(Line 793-802) "China totally absorbed the Soviet Union's planning concept of a huge industrial base and individual plant, which caused the formal sense to have more priority than the functional demands in determining the planning. In order to meet the needs of a longer assembly line, the huge workshops were built with a column net of 6 m x 18 m, which made it impossible to reserve a reasonable internal space for industrial upgrading right now. After almost 70 years, there are a large number of integrity problems on the facade, the roof, and the component, as well as the damage problems on the structure. The majority of the workshops in three cases became the industrial heritage since losing their manufacturing function. In order to conserve the unique sample of the industrial civilization and re-stimulate the urban economy, developing sustainable modern industrial tourism attracts our research interest."

(Line 803-813) "The Outline of China's Industrial Tourism Development (2016) released that 1000 national industrial tourism demonstration projects and 10 industrial tourism cities will be established in China until 2025. Changchun, Harbin, and Chengdu were all listed in the catalog. China will enter a golden development period of industrial tourism from 2020 to 2025. The total direct income will exceed 200 billion yuan, the number of new direct tourism employment will exceed 1.2 million, and the total comprehensive income for the regions and cities may exceed 10 times the direct income [67]. Through the statistics of bibliometrics, up to the end of 2019, more than 70 regions and cities in China have carried out relate studies on the development of industrial tourism projects, but the successful cases are still in lack because of the shortage of sustainable strategies [68]. We proposed two sustainable development strategies of industrial tourism based on the comprehensive assessment."

(Line 834-857) "Secondly, we established achievable and long-term goals and presented two reasonable patterns for the sustainable development of China's modern industrial tourism. We recommend the following: (1) The goal of informational spillover effect: the database of industrial heritage resources should be established, including the basic information, the GIS data, and assessment results of multi-value, which can be used as a reference for future conservation, regeneration and tourism development; (2) the goal of environmental and functional spillover effect: the industrial tourism projects should be equipped with ecological regulation capacity and more service content to improve their sustainability. On the one hand, we could expand the green garden areas and purify the water circulation system in the plants; on the other hand, we suggest adding the functions of technology exhibition, culture education, and leisure vacation into the above cases [69]; and (3) the goal of regional economic spillover effect: the comprehensive industrial tourism projects with special themes should be developed according to their unique characters and advantages. The Changchun First Automobile Works and the Harbin Boiler Factory are located on the important node cities, Changchun and Harbin, which are linked by the high-speed railway network. Both of the plants presented the achievement of China's heavy industry and exploration of modern industrial architecture. Accordingly, we could develop and design an urban linkage regional heavy industrial tourism project (Belt Pattern) based on the convenient railway, consisting of several supporting facilities located along the railway between the two cities. The Chengdu Hongguang Electronic Transistor Factory is located in the capital city of Sichuan Province, the center of China's electronic industry. We could develop and design a centralized electronic industry tourism project (Radiation Pattern), consisting of high technology enterprises, scientific research institutes, and outstanding universities around Chengdu, serving and supporting the main project. The above two patterns will stimulate the rapid development of regional economy and integration of advantageous industrial resources [70-72]."

Point 2: I wonder if there is more to be said on the representational aspect of industrial architecture and its framing as a place of labor relations. There is a paradoxical ethos: partly one of collective solidarity and at the same time individual alienation. If the factory was the place where that spirit was formerly spatialized, what has become of the significance of that ethos in today's industrial architecture? Is that ethos signified in any way in what the authors call "industrial tourism" today? What has happened to the social or cultural sustainability of that ethos today?

Response 2: We totally agree with your view that maintaining the sustainability of the culture and ethos in the industrial heritage is significant and meaningful content while developing industrial tourism. Helping tourists enjoy the workers' mental feelings will bring more spillover effect of the industrial tourism project. We have revised the following paragraph (Line 814-831) in new 4. Discussion (Page 29) for further discussion.

(Line 814-831) "Firstly, we calculated and counted the assessment scores for each workshop heritage based on the above assessments Table 3-5. We suggested reasonable reinforcing and repairing should be implemented according to the comprehensive assessment results. We recommend the following: (1) The workshops with high assessed score (more than 7) should be reinforced immediately, especially on their structure. We could increase the section area of the concrete columns and replace the broken components of the steel structure; (2) the workshops with middle assessed score (5-7) should be repaired on both facade and roof. We must respect their original aesthetic appearance and use original construction methods, building materials, and decoration color; and (3) the workshops with low assessed score (less than 5) should be minorly patched on parts of the walls, windows, stairs, and other components. The above three measures will serve the sustainability of architectural material space. In addition, we proposed that preserving the mechanical equipment and exhibiting the original manufacturing process would be necessary and beneficial for serving the sustainability of industrial culture. The industrial activities and workers' spiritual experience were the main content of the industrial culture. They were all spatialized by the equipment and assembly line in the architectural space. A successful and remarkable industrial tourism project should make tourists easily understand the manufacturing process and better enjoy the complicated mental feeling of self-identify brought by the industrial activities, collective solidarity stimulated by the cooperation, and individual alienation caused by the division of assembly line."

Point 3: Good images but some are low resolution. Figs 2, 9, 10, 12, 19, 31 all needing a resolution check. Formatting of references unusual. I can't seem to work this out, which is neither a bibliography nor a footnote. Please clarify with journal style guide for consistency.

Response 3: We have changed Figs 2, 9, 10, 12, 19, 32(original 30), 33 (original 31) with high resolution. We rescanned Figs 2, 9, 10 from the book: Hildebrand, G. Designing for Industry: Architecture of Albert Kahn; The MIT Press: Cambridge, UK, 1974. as well as improving the readability by the colors and mark words. We downloaded Figs 12 from Bentley Historical Library of University of Michigan and tried our best to increase its resolution. We made the mark words bigger and clearer in Figs 19, 32(original 30), 33 (original 31). Finally, we checked and clarified all the images in this manuscript by following the journal style guide. The footnotes of Figs 2, 3, 4, 5, 6, 7, 9, 10, 11, 12, 13, 14, 15, 16, 17, 18, 20 have been revised. Figs 1, 8, 19, 21, 22, 23, 24, 25, 26, 27, 28, 29, 30, 31, 32, 33 all come from our photos and works with copyrights.

Best regards,

Yours sincerely

Rui Han (on behalf of all authors)

School of Architecture, Harbin Institute of Technology , Harbin 150006, Heilongjiang, China.

Reviewer 2 Report

The paper outlines a study of the origin of China's modern industrial architecture and draws conclusions on sustainable development strategies and industrial tourism.

Overall this is an interesting article. However whilst the first part of the paper (literature review) is very good, the second part of the paper (methods, results, discussion, conclusion) is lacking in clarity, structure and sound scientific approach.

Whilst the start of the paper is inherently promising and based on a sound literature review, the actual methodology and results drawn from the analysis are relatively weak and inconclusive. The discussion and conclusion are mostly detached from the bulk of the paper, which is based on a sound analysis of the historical roots of the Chines industrial architecture. Subsequently the final conclusions are not supported by the methods applied and results shown.

The introduction as well as the following sections (Chapters 2, 3 and 4 on Literature Review of the US, Soviet Union and China) are very well written and researched. Only a few minor comments in this respect: Avoid subjective statements (line 88); avoid the term "indoor lighting" when you actually mean "daylighting" (lines 107, 144, 199, 807) as mostly indoor lighting is associated with artificial lighting. Correct a few typos (e.g. line 62, 188). 

The background sections runs until page 18, where the actual methodology (description of field investigation) begins. From this section onwards the paper becomes weak. The methodology interlinks with the background section as well as the results and the paper becomes less clear and very unstructured. Also, from this section onwards the English language deteriorates and should thus be thoroughly revised.

Section 4.4 presents the field investigation and measurements, however it is entangled with actual historical data as described in previous chapters. The two methods of field investigation (how, what, where) should be separated from historical data based on literature review. Also the methods on the interviews (page 23) must be more clearly described.

Chapter 5 is called discussion, but is actually a summary of the results and should thus be named as such. The results of the different methods (field investigation, interviews) are not clearly described and presented in a coherent manner.  From the current paper no clear results from the interviews can be determined.

It is further not conclusively stated, how the results as outlined in tables 2, 3 and 4 do relate to the methods applied. Also it is not explained on what basis the indexes were selected and on what key performance criteria the values were based on. The sustainable development strategies derived from these results are very far fetched, as there is no explanation what strategies currently exist and / or why there is a need for these strategies in this particular context.

Chapter 6 (conclusion) is actually more like a summary than a conclusion. This should be revised to provide the reader a sound assessment of the actual conclusions of the paper.

It is strongly recommended to fully revise the sections in methodologies, results and discussion to provide a scientifically sound assessment.

It is further recommended to structure the paper more clearly into: 1-introduction, 2-background (literature review, Chapters 2-4), 3-methodology (parts of Chapter 4) 4-results (parts of Chapter 5 and misleadingly named "Discussion) and 5-discussion (alternatively also 6-conclusion). 

Author Response

Dear reviewer,

Thank you for handling the review of our manuscript entitled “A Study on the Origin of China's Modern Industrial Architecture and Its Development Strategies of Industrial Tourism (Sustainability-775648)”. In this revised version, we have carefully addressed all the issues raised by you. We sincerely appreciate your insightful comments and suggestions that greatly help improve our manuscript.

The following is a summary of the point-to-point response and revisions we have made to each of your comments. We look forward to hearing your more suggestion and the outcome of this latest version of the manuscript.

Point 1: The paper outlines a study of the origin of China's modern industrial architecture and draws conclusions on sustainable development strategies and industrial tourism. Overall this is an interesting article. However whilst the first part of the paper (literature review) is very good, the second part of the paper (methods, results, discussion, conclusion) is lacking in clarity, structure and sound scientific approach. Whilst the start of the paper is inherently promising and based on a sound literature review, the actual methodology and results drawn from the analysis are relatively weak and inconclusive. The discussion and conclusion are mostly detached from the bulk of the paper, which is based on a sound analysis of the historical roots of the Chinese industrial architecture. Subsequently the final conclusions are not supported by the methods applied and results shown.

Response 1: Thank you for the suggestion which benefits us a lot. We have made a moderate revision for the structure of the manuscript following your advice, and tried to make the methodology, discussion, and conclusion clearer. The new version is as follows: 1. Introduction; 2. Background (we described the development of the modern industrial architecture in the United States (section 2.1), the Soviet Union (section 2.3), and China (section 2.4) with a literature review.); 3. Methods (field investigation: the inheritance and changes of modern industrial architecture in China (section 3.1), measurement: the integrity and damage of the modern industrial heritage (section 3.2), and face-to-face interview: the multi-value of the modern industrial heritage (section 3.3).); 4. Discussion (we clarified the origin of China's modern industrial architecture, analyzed the dilemma of conservation and regeneration, and proposed sustainable development strategies for the industrial heritage); and 5. Conclusion.

Firstly, in order to improve the continuity of the literature review and achieve a reliable conclusion, we integrated the content of the development of modern industrial architecture in the United States, the Soviet Union, and China together into the new section: 2. Background, and make it better serve the new section: 3. Methods. We added more sound scientific approach in analysis and assessment. We have revised the following paragraph (Line 632-640, Line 753-762, and Line 767-776), and added new Table 2 (Line 764), new Figure 30,31 (Line 777-784) for further discussion.

(Line 632-640) "Jilin province, Heilongjiang province, and Sichuan province are the most important industrial provinces in China in the 1950s. A lot of core industrial plants were built in the capital cities of these provinces. The Changchun First Automobile Works, the Harbin Boiler Factory, and the Chengdu Hongguang Electronic Transistor Factory were selected as the study cases since their distinctive features. They all represented the outstanding achievement of China's modern industrial architecture, which helps us easily observe the inheritance and compare the changes among the United States, the Soviet Union, and China from the perspective of the planning concept, design theory, and structural technology. In addition, the industrial heritages in these cases are all facing the difficulties of conservation and regeneration. A deep field investigation is urgent and necessary."

(Line 753-762) "We established a statistical table of integrity and damage based on the record of the measurement. The integrity problems mainly included facade integrity, roof integrity, and component integrity. We measured all the 13 samples in three cases, recorded 608 facade integrity problems, 113 roof integrity problems, and 602 building component integrity problems. The damage problems referred to the structure damage. We recorded 156 structure damage problems, 44 of which were serious problems, mainly emerged on reinforced concrete columns. According to the statistics, we established a general assessment for future conservation and regeneration. There are three levels in the general assessment as follows: (1) Worse level with 3 scores: more than 140 integrity and damage problems; (2) normal level with 2 scores: 90-140 integrity and damage problems; and (3) better level with 1 score: less than 90 integrity and damage problems."

(Line 767-774) "We conducted face-to-face interview for multi-value such as historical, economic, aesthetic, and technological value at the end of 2019. 137 people working in the Changchun First Automobile Works, 115 people working in the Harbin Boiler Factory, and 122 people working in the Chengdu Hongguang Electronic Transistor Factory, were surveyed. About 35% of them were 45-55 years old and employed more than 25 years; 30% of them were 35-44 years old, and employed more than 15 years; 35% of them were 20-34 years old, and employed more than 10 years. They all witnessed the historical changes of these three plants in different eras, and presented relatively objective assessment according to their different ages, world outlook, and aesthetic taste."

Secondly, based on the new structure, we made a clearer conclusion of the origin of China's modern industrial architecture according to the literature review and the field investigation. In addition, we presented a reasonable background and sustainable development strategies of industrial tourism for the cases in the new section: 4. Discussion according to the measurement, face-to-face interview, and assessments. we have revised the following paragraph (Line 793-802, Line 814-831, and Line 834-857), and added the following paragraph (Line 803-813) in new 4. Discussion (Page 28-30).

(Line 793-802) "China totally absorbed the Soviet Union's planning concept of a huge industrial base and individual plant, which caused the formal sense to have more priority than the functional demands in determining the planning. In order to meet the needs of a longer assembly line, the huge workshops were built with a column net of 6 m x 18 m, which made it impossible to reserve a reasonable internal space for industrial upgrading right now. After almost 70 years, there are a large number of integrity problems on the facade, the roof, and the component, as well as the damage problems on the structure. The majority of the workshops in three cases became the industrial heritage since losing their manufacturing function. In order to conserve the unique sample of the industrial civilization and re-stimulate the urban economy, developing sustainable modern industrial tourism attracts our research interest."

(Line 803-813) "The Outline of China's Industrial Tourism Development (2016) released that 1000 national industrial tourism demonstration projects and 10 industrial tourism cities will be established in China until 2025. Changchun, Harbin, and Chengdu were all listed in the catalog. China will enter a golden development period of industrial tourism from 2020 to 2025. The total direct income will exceed 200 billion yuan, the number of new direct tourism employment will exceed 1.2 million, and the total comprehensive income for the regions and cities may exceed 10 times the direct income [67]. Through the statistics of bibliometrics, up to the end of 2019, more than 70 regions and cities in China have carried out relate studies on the development of industrial tourism projects, but the successful cases are still in lack because of the shortage of sustainable strategies [68]. We proposed two sustainable development strategies of industrial tourism based on the comprehensive assessment."

(Line 814-831) "Firstly, we calculated and counted the assessment scores for each workshop heritage based on the above assessments Table 3-5. We suggested reasonable reinforcing and repairing should be implemented according to the comprehensive assessment results. We recommend the following: (1) The workshops with high assessed score (more than 7) should be reinforced immediately, especially on their structure. We could increase the section area of the concrete columns and replace the broken components of the steel structure; (2) the workshops with middle assessed score (5-7) should be repaired on both facade and roof. We must respect their original aesthetic appearance and use original construction methods, building materials, and decoration color; and (3) the workshops with low assessed score (less than 5) should be minorly patched on parts of the walls, windows, stairs, and other components. The above three measures will serve the sustainability of architectural material space. In addition, we proposed that preserving the mechanical equipment and exhibiting the original manufacturing process would be necessary and beneficial for serving the sustainability of industrial culture. The industrial activities and workers' spiritual experience were the main content of the industrial culture. They were all spatialized by the equipment and assembly line in the architectural space. A successful and remarkable industrial tourism project should make tourists easily understand the manufacturing process and better enjoy the complicated mental feeling of self-identify brought by the industrial activities, collective solidarity stimulated by the cooperation, and individual alienation caused by the division of assembly line."

(Line 834-857) "Secondly, we established achievable and long-term goals and presented two reasonable patterns for the sustainable development of China's modern industrial tourism. We recommend the following: (1) The goal of informational spillover effect: the database of industrial heritage resources should be established, including the basic information, the GIS data, and assessment results of multi-value, which can be used as a reference for future conservation, regeneration and tourism development; (2) the goal of environmental and functional spillover effect: the industrial tourism projects should be equipped with ecological regulation capacity and more service content to improve their sustainability. On the one hand, we could expand the green garden areas and purify the water circulation system in the plants; on the other hand, we suggest adding the functions of technology exhibition, culture education, and leisure vacation into the above cases [69]; and (3) the goal of regional economic spillover effect: the comprehensive industrial tourism projects with special themes should be developed according to their unique characters and advantages. The Changchun First Automobile Works and the Harbin Boiler Factory are located on the important node cities, Changchun and Harbin, which are linked by the high-speed railway network. Both of the plants presented the achievement of China's heavy industry and exploration of modern industrial architecture. Accordingly, we could develop and design an urban linkage regional heavy industrial tourism project (Belt Pattern) based on the convenient railway, consisting of several supporting facilities located along the railway between the two cities. The Chengdu Hongguang Electronic Transistor Factory is located in the capital city of Sichuan Province, the center of China's electronic industry. We could develop and design a centralized electronic industry tourism project (Radiation Pattern), consisting of high technology enterprises, scientific research institutes, and outstanding universities around Chengdu, serving and supporting the main project. The above two patterns will stimulate the rapid development of regional economy and integration of advantageous industrial resources [70-72]."

Point 2: The introduction as well as the following sections (Chapters 2, 3 and 4 on Literature Review of the US, Soviet Union and China) are very well written and researched. Only a few minor comments in this respect: Avoid subjective statements (line 88); avoid the term "indoor lighting" when you actually mean "day lighting" (lines 107, 144, 199, 807) as mostly indoor lighting is associated with artificial lighting. Correct a few typos (e.g. line 62, 188).

Response 2: The discussion and review of this paragraph (line 76-91) actually benefited a lot from the reference [5] (Bradley, B. The Works: The Industrial Architecture of the United States; Oxford University Publishing: New York, US, 1999). We checked carefully and revised the position of reference [5] and put it at the end of this paragraph (line 91) to avoid subjective statements. We changed the term "indoor lighting" by "day lighting" (Line 109, 147, 202, 223, 233, 257, 374) and corrected the errors (Line 62, 191) and the other grammar errors after checking the whole manuscript carefully.

Point 3: The background sections runs until page 18, where the actual methodology (description of field investigation) begins. From this section onwards the paper becomes weak. The methodology interlinks with the background section as well as the results and the paper becomes less clear and very unstructured. Also, from this section onwards the English language deteriorates and should thus be thoroughly revised.

Response 3: We revised the structure of both the whole manuscript and every section, added sound analysis and scientific approach, and formed the new section: 2. Background, 3. Methods, 4. Discussion. In the new section: 3. Methods, we described the field investigation, measurement, and face-to-face interview more technically, and added more data to explain the assessment. The adding content will be described in the following Responses 4, 6. We also have checked the grammar in the whole manuscript carefully and corrected the errors.

Point 4: Section 4.4 presents the field investigation and measurements, however it is entangled with actual historical data as described in previous chapters. The two methods of field investigation (how, what, where) should be separated from historical data based on literature review. Also the methods on the interviews (page 23) must be more clearly described.

Response 4: We divided the origin Section 4 into two parts. The first part of the literature review was put into the new section: 2. Background. We integrated second part of the field investigation, the measurement, the face-to-face interview, the statistics of integrity and damage, and the multi-value assessment together into the new section: 3. Methods. we have revised the following paragraph (Line 767-776) and added new Figure 30, 31 (777, 781) to describe the face-to-face interview more clearly.

(Line 767-776) "We conducted face-to-face interview for multi-value such as historical, economic, aesthetic, and technological value at the end of 2019. 137 people working in the Changchun First Automobile Works, 115 people working in the Harbin Boiler Factory, and 122 people working in the Chengdu Hongguang Electronic Transistor Factory, were surveyed. About 35% of them were 45-55 years old and employed more than 25 years; 30% of them were 35-44 years old, and employed more than 15 years; 35% of them were 20-34 years old, and employed more than 10 years. They all witnessed the historical changes of these three plants in different eras, and presented relatively objective assessment according to their different ages, world outlook, and aesthetic taste. The interview analysis of the historical and economic value is shown in Figure 30. The interview records of aesthetic and technological value is shown in Figure 31."

Point 5: Chapter 5 is called discussion, but is actually a summary of the results and should thus be named as such. The results of the different methods (field investigation, interviews) are not clearly described and presented in a coherent manner. From the current paper no clear results from the interviews can be determined.

Response 5: We have revised the origin section 5 and formed a new section: 4.Discussion. Firstly, we made a brief discussion on the origin of China's modern industrial architecture based on the literature review; secondly, we analyzed the difficulties faced by industrial heritage based on the measurement and interview; Thirdly, we proposed sustainable development strategies based on policy of industrial tourism. The adding content will be described in the following Response 7.

Point 6: It is further not conclusively stated, how the results as outlined in tables 2, 3 and 4 do relate to the methods applied. Also it is not explained on what basis the indexes were selected and on what key performance criteria the values were based on.

Response 6: Firstly, we have revised the following paragraph (Line 767-776 and Line 785-789) and added new Figure 30, 31 (777, 781) to explain the assessment result of new Table 4 (original Table 2, Line 790) and new Table 5 (original Table 3, Line 791).

(Line 767-776) "We conducted face-to-face interview for multi-value such as historical, economic, aesthetic, and technological value at the end of 2019. 137 people working in the Changchun First Automobile Works, 115 people working in the Harbin Boiler Factory, and 122 people working in the Chengdu Hongguang Electronic Transistor Factory, were surveyed. About 35% of them were 45-55 years old and employed more than 25 years; 30% of them were 35-44 years old, and employed more than 15 years; 35% of them were 20-34 years old, and employed more than 10 years. They all witnessed the historical changes of these three plants in different eras, and presented relatively objective assessment according to their different ages, world outlook, and aesthetic taste. The interview analysis of the historical and economic value is shown in Figure 30. The interview records of aesthetic and technological value is shown in Figure 31."

(Line 785-789) "According to the statistics of the interview records, we established the assessment of historical and economic value and aesthetic and technological value. Different scores (1-3) present different value level objectively, which provides us a reference for conservation and regeneration. The assessment of historical and economic value is listed in Table 4. The assessment of aesthetic and technological value is listed in Table 5."

Secondly, we have revised the following paragraph (Line 753-763) and added new Table 2 (764) to explain the assessment result of new Table 3 (original Table 4, Line 765).

(Line 753-763) "We established a statistical table of integrity and damage based on the record of the measurement. The integrity problems mainly included facade integrity, roof integrity, and component integrity. We measured all the 13 samples in three cases, recorded 608 facade integrity problems, 113 roof integrity problems, and 602 building component integrity problems. The damage problems referred to the structure damage. We recorded 156 structure damage problems, 44 of which were serious problems, mainly emerged on reinforced concrete columns. According to the statistics, we established a general assessment for future conservation and regeneration. There are three levels in the general assessment as follows: (1) Worse level with 3 scores: more than 140 integrity and damage problems; (2) normal level with 2 scores: 90-140 integrity and damage problems; and (3) better level with 1 score: less than 90 integrity and damage problems. The statistics of integrity and damage is listed in Table 2."

Table 2. The statistics of integrity and damage of the 13 samples.

Case

Workshop

Integrity Problem

Facade   Roof  Component

Structure Damage

Normal  Serious

Changchun First Automobile Works

Punching Press Workshop

48

6

37

9

1

Cutting Workshop

38

11

32

11

2

Foundry Workshop

68

14

77

16

9

Welding Workshop

59

7

66

15

11

Assembly Workshop

70

19

80

9

7

Power Station

45

10

35

12

2

Harbin Boiler Factory

Cutting Workshop

42

9

30

4

2

Foundry Workshop

55

10

49

2

0

Welding Workshop

43

3

31

6

2

Assembly Workshop

51

7

72

8

1

Chengdu Hongguang Electronic Transistor Factory

Office Building

22

5

25

5

2

Testing Workshop

27

5

29

9

1

Assembly Workshop

40

7

39

6

4

Point 7: The sustainable development strategies derived from these results are very far fetched, as there is no explanation what strategies currently exist and / or why there is a need for these strategies in this particular context.

Response 7: In order to explain what strategies currently exist and why there is a need for these strategies in the new section: 4.Discussion. We have revised the following paragraph (Line 793-802, Line 803-813, Line 814-832, and Line 834-859).

(Line 793-802) "China totally absorbed the Soviet Union's planning concept of a huge industrial base and individual plant, which caused the formal sense to have more priority than the functional demands in determining the planning. In order to meet the needs of a longer assembly line, the huge workshops were built with a column net of 6 m x 18 m, which made it impossible to reserve a reasonable internal space for industrial upgrading right now. After almost 70 years, there are a large number of integrity problems on the facade, the roof, and the component, as well as the damage problems on the structure. The majority of the workshops in three cases became the industrial heritage since losing their manufacturing function. In order to conserve the unique sample of the industrial civilization and re-stimulate the urban economy, developing sustainable modern industrial tourism attracts our research interest."

(Line 803-813) "The Outline of China's Industrial Tourism Development (2016) released that 1000 national industrial tourism demonstration projects and 10 industrial tourism cities will be established in China until 2025. Changchun, Harbin, and Chengdu were all listed in the catalog. China will enter a golden development period of industrial tourism from 2020 to 2025. The total direct income will exceed 200 billion yuan, the number of new direct tourism employment will exceed 1.2 million, and the total comprehensive income for the regions and cities may exceed 10 times the direct income [67]. Through the statistics of bibliometrics, up to the end of 2019, more than 70 regions and cities in China have carried out relate studies on the development of industrial tourism projects, but the successful cases are still in lack because of the shortage of sustainable strategies [68]. We proposed two sustainable development strategies of industrial tourism based on the comprehensive assessment."

(Line 814-832) "Firstly, we calculated and counted the assessment scores for each workshop heritage based on the above assessments Table 3-5. We suggested reasonable reinforcing and repairing should be implemented according to the comprehensive assessment results. We recommend the following: (1) The workshops with high assessed score (more than 7) should be reinforced immediately, especially on their structure. We could increase the section area of the concrete columns and replace the broken components of the steel structure; (2) the workshops with middle assessed score (5-7) should be repaired on both facade and roof. We must respect their original aesthetic appearance and use original construction methods, building materials, and decoration color; and (3) the workshops with low assessed score (less than 5) should be minorly patched on parts of the walls, windows, stairs, and other components. The above three measures will serve the sustainability of architectural material space. In addition, we proposed that preserving the mechanical equipment and exhibiting the original manufacturing process would be necessary and beneficial for serving the sustainability of industrial culture. The industrial activities and workers' spiritual experience were the main content of the industrial culture. They were all spatialized by the equipment and assembly line in the architectural space. A successful and remarkable industrial tourism project should make tourists easily understand the manufacturing process and better enjoy the complicated mental feeling of self-identify brought by the industrial activities, collective solidarity stimulated by the cooperation, and individual alienation caused by the division of assembly line."

(Line 834-859) "Secondly, we established achievable and long-term goals and presented two reasonable patterns for the sustainable development of China's modern industrial tourism. We recommend the following: (1) The goal of informational spillover effect: the database of industrial heritage resources should be established, including the basic information, the GIS data, and assessment results of multi-value, which can be used as a reference for future conservation, regeneration and tourism development; (2) the goal of environmental and functional spillover effect: the industrial tourism projects should be equipped with ecological regulation capacity and more service content to improve their sustainability. On the one hand, we could expand the green garden areas and purify the water circulation system in the plants; on the other hand, we suggest adding the functions of technology exhibition, culture education, and leisure vacation into the above cases [69]; and (3) the goal of regional economic spillover effect: the comprehensive industrial tourism projects with special themes should be developed according to their unique characters and advantages. The Changchun First Automobile Works and the Harbin Boiler Factory are located on the important node cities, Changchun and Harbin, which are linked by the high-speed railway network. Both of the plants presented the achievement of China's heavy industry and exploration of modern industrial architecture. Accordingly, we could develop and design an urban linkage regional heavy industrial tourism project (Belt Pattern) based on the convenient railway, consisting of several supporting facilities located along the railway between the two cities. The Chengdu Hongguang Electronic Transistor Factory is located in the capital city of Sichuan Province, the center of China's electronic industry. We could develop and design a centralized electronic industry tourism project (Radiation Pattern), consisting of high technology enterprises, scientific research institutes, and outstanding universities around Chengdu, serving and supporting the main project. The above two patterns will stimulate the rapid development of regional economy and integration of advantageous industrial resources [70-72]."

Point 8: Chapter 6 (conclusion) is actually more like a summary than a conclusion. This should be revised to provide the reader a sound assessment of the actual conclusions of the paper. It is strongly recommended to fully revise the sections in methodologies, results and discussion to provide a scientifically sound assessment.

Response 8: We have revised the following paragraph (Line 867-934) in new 5. Conclusion (Page 32,33).

(Line 867-934) "There existed a clear route of the global transfer of modern industrial architecture from the United States to the Soviet Union and then to China in the first half of the 20th century. The origin of China's modern industrial architecture came from the achievement of the United States' modern industrial architecture, meanwhile adding the Soviet Union's standard construction technology. Both of them were integrated into a new industrial architectural vision by the prevailing ideology in China in the 1950s.

Modern industrial architecture, founded and perfected by the American architects, was imported into the Soviet Union during its First Five-Year Plan (1928-1932) through the channel of directing projects and training architects. There was no phenomenon of exclusion in the process of the transfer of modern industrial architecture from the United States to the Soviet Union. In China's First Five-Year Plan (1953-1957), the Soviet Union's modern industrial architecture, was exported to China through the assistance of the "156 Projects". The transfer channel included sending experts, directing projects, and interpreting books. In order to meet the needs of the Chinese domestic political culture and express the prevailing ideological trend of architectural design, a unique style of the modern industrial architecture was invented. The facade vision of the workshops expressed the Chinese national form, which became a precious and valuable sample of the development process of human industrial civilization. At the end of the 1950s, the decoration of China's modern industrial architecture was simplified and began to display a clearer modernist style which was more closed to the United States.

China's modern industrial architecture have high multi-value. Besides, the problems of integrity and damage were extremely obvious. We found that 1210 integrity problems (81.8 % of the total) emerged on the facade and roof since the aging of building materials, and 156 damage problems (10.5 % of the total) emerged on the reinforced concrete columns since the concrete damage. The majority of the workshops became the industrial heritages which are facing the dilemma of conservation and regeneration. Based on the comprehensive assessment, we proposed two sustainable development strategies and two patterns for the industrial tourism projects as following:

In order to improve the sustainability of both architectural material space and industrial culture, we should classify the industrial heritages and implement different measures of repairing and reinforcing, meanwhile preserving the mechanical equipment and exhibiting the original manufacturing process.

A database of industrial heritage resources should be established for future reference and management. Equipping with more service content and ecological regulation capacity could realize the goal of environmental and functional spillover effect. We presented "Belt Pattern" and "Radiation Pattern" for the tourism development of the cases, which depend on their different themes and respective advantages."

Point 9: It is further recommended to structure the paper more clearly into: 1-introduction, 2-background (literature review, Chapters 2-4), 3-methodology (parts of Chapter 4) 4-results (parts of Chapter 5 and misleadingly named "Discussion) and 5-discussion (alternatively also 6-conclusion).

Response 9: We do appreciate for your suggestion which helps us perfect the structure of the manuscript. We have revised the structure following your advice in the new version.

Best regards,

Yours sincerely

Rui Han (on behalf of all authors)

School of Architecture, Harbin Institute of Technology , Harbin 150006, Heilongjiang, China.

Round 2

Reviewer 2 Report

Thank you for the revised manuscript. The paper has now significantly improved. The structure is much clearer and the overall aim of the paper has been much better documented. 

A few minor additional comments: 

Section 3 Methods would benefit from a short introduction to guide the reader into the section, e.g. why have you chosen these methods? Why are these methods particularly suitable for your research?

Generally the Methods are still interlinked with the Results. As you do not have a separate section for the Results I would recommend to make a subheading within your 3 methods to clearly differentiate from the description of the method and the results achieved with the method. 

There seems to be a missing text in Line 778. 

Check again grammar and English context in the discussion and conclusion. e.g. the last sentence "..Equipping with more services...." is not proper English. Since the last two chapters are those that are probably read the most, I would strongly suggest another review of the text. 

In conclusion however I think this in an interesting paper, which now also reads well. 

Author Response

Dear reviewer,

Thank you very much for handling the review of our manuscript. In this revised version, we have carefully addressed all the issues raised by you. We sincerely appreciate your insightful comments and suggestions that greatly help improve our manuscript.

The following is a summary of the point-to-point response and revisions we have made to each of your comments.

Point 1: Thank you for the revised manuscript. The paper has now significantly improved. The structure is much clearer and the overall aim of the paper has been much better documented. A few minor additional comments: Section 3 Methods would benefit from a short introduction to guide the reader into the section, e.g. why have you chosen these methods? Why are these methods particularly suitable for your research?

Response 1: Thank you for the suggestion. We have added the following paragraph (Line 602-610) at the beginning of Section 3 for a short introduction, and explained the purpose and reason why choosing these methods.

(Line 602-610) "Base on the previous literature review, we described the global transfer of modern industrial architecture from the United States to the Soviet Union and then to China. For the purpose of further proving the relevance of the modern industrial architecture among the three countries and finding the inheritance and changes existing in the "156 Projects". we were determined to conduct field investigation immediately. In this process, we also implemented careful measurement and face-to-face interview to understand the status of the workshops and achieve the objective assessment of their multi-value. The results would not only serve the evidence to prove the origin of China's modern industrial architecture but also benefit the conservation and regeneration of them in the future."

Point 2: Generally the Methods are still interlinked with the Results. As you do not have a separate section for the Results I would recommend to make a subheading within your 3 methods to clearly differentiate from the description of the method and the results achieved with the method.

Response 2: We have added the 3.4. results (Line 765-774) at the end of Section 3 for making a brief summary and a shot introduction for the discussion of the development of industrial tourism.

(Line 765-774) "Due to facing the dilemma of conservation and regeneration, more than 80% of the workshops have become industrial heritage. A large number of integrity and damage problems were appearing, 41.2% of which emerged on the facade, 7.6% of which emerged on the roof, 40.7% of which emerged on the components, and 10.5% of which emerged on the structure. The statistics revealed that these industrial heritages have good structural stability, which proved the necessity of conservation and the possibility of regeneration. Base on a face-to-face interview, we have confirmed that nearly 40% of them have a high historical and economic value and nearly 60% of them have a high aesthetic and technological value. As a result, how to better implement conservation and regeneration became the key."

Point 3: There seems to be a missing text in Line 778.Check again grammar and English context in the discussion and conclusion. e.g. the last sentence "..Equipping with more services...." is not proper English. Since the last two chapters are those that are probably read the most, I would strongly suggest another review of the text. In conclusion however I think this in an interesting paper, which now also reads well.

Response 3: We have checked the grammar in the whole manuscript carefully and corrected all the errors to make a clearer presentation. Thank you again for your insightful suggestions. After this revision, this manuscript has a better structure, continuity, and readability. We have addressed all the issues raised by you and revised the words and phrases as follows:(Line 214, 230, 292, 331, 373, 384,586, 615, 626,797, 823, 880).

Best regards,

Yours sincerely

Rui Han (on behalf of all authors)

School of Architecture, Harbin Institute of Technology , Harbin 150006, Heilongjiang, China.
